# Leaf Epidermal Traits of Selected Euphorbiaceae and Phyllanthaceae Taxa of Hainan Island and Their Taxonomic Relevance

**Saraj Bahadur [1], Mushtaq Ahmad [2], Wenxing Long [1,*], Muhammad Yaseen [1] and Uzma Hanif [3]**

[1] Wuzhishan National Long-Term Forest Ecosystem Monitoring Research Station, College of Forestry, Hainan University, Haikou 570228, China
[2] Department of Plant Sciences, Quaid-i-Azam University, Islamabad 45320, Pakistan
[3] Department of Botany, Government College University, Lahore 54000, Pakistan
* Correspondence: oklong@hainanu.edu.cn

**Abstract:** Euphorbiaceae and Phyllantaceae are two of the most taxonomically complex and diverse families among angiosperm. Therefore, this study aims to identify the common, variation, and diagnostic traits of foliar anatomy and micromorphology of selected Euphorbiaceae and Phyllanthaceae taxa collected from Hainan Island. A total of 42 individuals of 38 species belonging to these two families were examined under light microscopy (LM) and scanning electron microscopy (SEM). A multivariate detrended correspondence analysis (DCA), principal component analysis (PCA), and chord diagram analysis were performed to visualize the variations and relationships between the species. The nonglandular trichomes were further classified into long falcate and short conical with papillae on their surface, and stellate and lepidote subentire, while the glandular trichomes were divided into capitate, subsessile capitate, sessile capitates, and peltate. The paracytic-type stomata were found in most of the species. The prominent continuous or discontinuous cuticular striations radiating from the guard cells in parallel, horizontal, or reticulate patterns were analyzed in most of the taxa. Variation was also found in epidermal cells and anticlinal wall patterns. Together, these traits are useful for identification, discrimination, and to define species boundaries at various levels. Our study will provide a basis for future studies to strengthen the systematic of Euphorbiaceae taxa.

**Keywords:** Euphorbiaceae; Phyllantaceae; foliar epidermal anatomy; taxonomic significance; light and scanning electron microscopy; statistical analysis

## 1. Introduction

The family Euphorbiaceae is the sixth largest family comprising 322 genera and 8910 species occupying varying habitats, most commonly in subtropical and humid tropical regions of both hemispheres [1–3]. The family is among the top 25 economically important plant families and includes so commercially valuable taxa, such as *Aleurites*, *Hevea*, *Ricinus communis*, and *Manihot* [4]. The family is known for its ornamental taxa as well, such as *Euphorbia*, *Croton*, and *Acalypha*. In addition, it is ethnomedicinally important as well. In ancient Chinese traditional medicine, about 33 species belonging to 17 genera of Euphorbiaceae are reported to be used in herbal medicines [5]. In China, there are about 67 genera and 406 species of the family Euphorbiaceae [6].

A significant contribution to the family Euphorbiaceae was carried out by Webster, [7] and it was divided into five subfamilies (Phyllanthoideae, Oldfieldioideae, Acalyphoideae, Crotonoideae, and Euphorbioideae) based on the various morphological traits, such as the number of ovules per locule, occurrence of latex, indumentum type, and pollen morphology. Recently, the Angiosperm Phylogeny Group (APG IV) [8] based on the work of Wurdack et al. [9] recognized five lineages of Euphorbiaceae at the family level

(Phyllanthaceae, Putranjivaceae, Pandaceae, Picrodendraceae, and Euphorbiaceae) senso strict. The classification of these families is mainly based on the most advanced molecular taxonomy in conjunction with the morphological data proposed by Webster [7]. Most recent studies [10–18], confirmed the newly recognized APG families concerning molecular studies using DNA sequencing data as well as morphological traits. The first comprehensive study of Phyllanthaceae as one of the five segregates of Euphorbiaceae sl was published by Wurdack et al. [9]. Furthermore, the delimitation of Phyllanthaceae was early supported by several studies based on various aspects; seed coat [19], seed protein and serological data [20,21], chemotaxonomy [22], and phytochemistry [23]. Recently, Moawed et al. [24], recommended the segregation of Phyllanthaceae from Euphorbiaceae sl based on the micro- and macro-morphological as well as architectural characters. Hence further studies are needed to support the individual family status of these two families.

In the last few decades, a remarkable evolution has occurred in the anatomical study of vascular plants and their application in the taxonomy [25,26]. The family is well known for its rich diversity in floral, vegetative, anatomical, embryological, palynological, chemical, and cytological features [27,28]. In the structure of the leaf, the stomatal apparatus is not uniform [29]. One type of stomata is not widespread in the family and their distribution may also vary significantly [30]. The structural detail of leaf epidermis in Euphorbiaceae taxa except Euphorbia is rare when compared to the previously published literature. Most of the information refers to the tribes Acalypheae and Euphorbieae. Morphological and phylogenetic studies have helped to clarify the systematic of complex groups (i.e., tribe Crotoneae). However, very little is known about the structure, function, and diversity of the anatomical features in this tribe using scanning electron microscopy (SEM) [31]. Moreover, SEM is not only used to explore the fine detail of Euphorbiaceae taxa but in other groups of plants as well [32–37].

Few leaf epidermal studies of the family Euphorbiaceae have been documented in different geographical regions of the world over time. For example, the leaf anatomical studies of some economically important Euphorbiaceae taxa which have been studied previously such as the introduced species *Manihot ultissima* and *Hevea brasiliensis* [38,39] were confined to only some aspects (i.e., stomatal density, functioning of planting conditions, and canopy type). The taxonomic significance of the foliar epidermis in selected members of the Euphorbiaceae in Nigeria has been investigated previously [40]. While the root anatomy of medicinally important Euphorbiaceae taxa has also been investigated for quality assessment in the herbal industry [41]. Various foliar epidermis and foliar stomata of 17 species belonging to 17 genera of the Euphorbiaceae were analyzed in India. These anatomical features were found useful to reach the taxonomic and phylogenetic hypothesis [42]. The structure of foliar stomata of 50 Euphorbiaceae taxa belonging to 28 genera and 17 tribes were documented previously [27]. The foliar anatomical studies of 15 Euphorbiaceae taxa belonging to 15 genera were investigated to understand the structural details in unraveling taxonomic disputes [43]. The general anatomical studies of the family Euphorbiaceae, reveal that there are data about the mature stomata only and none on stomatogenesis [11]. He further stated that sometimes mixed types of stomata such as anisocytic and paracytic, occurred in a single leaf of Euphorbiaceae. A range of characters varies between genera and species, while a combination of these characters may be used to correctly identify the species and have significant potential.

Epidermal traits are chosen because they are mildly influenced by environmental factors and have a rich diversity in structure [44]. Similarly, some leaf epidermal studies are documented at the generic level of Euphorbiaceae. For example, light microscopy (LM) and scanning electron microscopic (SEM) studies of the stomata, mostly in *Euphorbia* and some other taxa of Euphorbiaceae showed various types of epicuticular wax coats and patterns of stomata closing their structure [45,46]. The leaf anatomical studies of the genus *Euphorbia* showed more than one type of stomata [47–49] and also showed that anatomical traits (i.e., epidermal cell shape, anticlinal wall pattern, and presence or absence of trichomes) had a taxonomic value and are useful to define species boundaries for correct identification [49]. The epidermal surface of the genus *Acalypha* is analyzed under

SEM and various types of crystals are observed and should be considered as a potential taxonomic traits [50]. However, further studies exploring the taxonomic sampling of this genus are still needed. To differentiate the two *Croton* species, a morphological and anatomical study was carried out by Bhavana et al. [51]. The genus *Aleurites* are characterized by the presence of curved cuticular ridges of peristomatal rim and a single outer stomatal ledge [52]. Comparative epidermal and petiole morphology and their taxonomic significance in the identification and elucidation of species affinity have been considered for the West African *Jatropha* by using both LM and SEM techniques [53]. Variations in the surface patterns of cuticles have attracted the attention of botanists. The reticulate cuticle on the foliar epidermis in *Hevea* was observed under both LM and SEM and demonstrated various types of cuticles (i.e., reticulate, lamellate, and striate) [54]. The leaf anatomy of *Ricinus communis* was investigated by Mamoucha et al. [55] and anisocytic stomata with unicellular nonglandular trichomes were reported. Similarly, the environmental impact on the morphology and anatomy of this species were analyzed by Suwal et al. [56]. The stomatal distribution and leaf thickness of *Codiaeum variegatum* and *Excoecaria cochinchinensis* have been studied previously [57]. However, no foliar epidermal studies of the family Euphorbiaceae and Phyllanthaceae are yet documented in Hainan Island South China. Hence, further studies are needed to fill this anatomical gap and provide useful anatomical traits and their relevance in the taxonomy of these two families.

Keeping in mind the significance of leaf epidermal traits, we designed the present with an aim to; (1) to explore the diversity in the leaf epidermal anatomy of Euphorbiaceae and Phyllanthaceae taxa; and (2) to analyze the leaf epidermal traits through light and scanning electron microscopy and their relevance in the taxonomy of this family collected from different geographical regions of the Hainan Island.

## 2. Materials and Methods

### 2.1. Study Area

Hainan Island is the second largest island in China with rich plant diversity and lies at 108°36′43″ E and 18°10′04″–20°9′40″ N. The area of this island is 33,920 km$^2$ with lowlands at the periphery and high mountains in the center. Although the island has abundant rainfall, it is suffering from water scarcity. This situation is attributed to several causes. For example, rainfall distributes unevenly in time and space [58–60]. The island has a relatively wet climate in the east with 2000 to 2400 mm annual rainfall and a dry climate in the west with 1000 to 1200 mm. Furthermore, November to May is the dry season and June to October is the rainy season with 70 to 90% of the annual rainfall [61–63].

### 2.2. Collection and Identification

In March 2021, several field trips were arranged to various parts of Hainan Island (Table 1). A total of 42 individuals of 38 species belonging to three triovulate subfamilies of Euphorbiaceae and the family Phyllanthaceae were randomly collected. The plant species were first identified with the help of already available specimens in the Herbarium of Hainan University and were then confirmed by using the available literature, Flora of China (Flora of China@eflora.org).

**Table 1.** Euphorbiaceae and Phyllanthaceae taxa from the study area of Hainan Island.

| Species | Locality | Voucher Number | Latitude | Longitude | Meter above Sea Level |
|---|---|---|---|---|---|
| **Subfamily Acalyphoideae** | | | | | |
| *Acalypha hispida* Burm. f. | Haikou | SB-HN-101 | 20.062363 | 110.3186713 | 3 |
| *Acalypha indica* L. | Chang Jiang | SB-HN-110 | 19.33905 | 108.210 | 33 |
| *Acalypha wilkesiana* Müll. Arg. | Haikou, **Wanning** | SB-HN-115 | 18.69396 | 110.23170 | 118 |
| *Alchornea rugosa* (Lour.) Müll. Arg. | Wanning, Botanical Garden | SB-HN-131 | 18.69396 | 110.23170 | 90 |

| | | | | | |
|---|---|---|---|---|---|
| *Macaranga corymbosa* Müll. Arg. | | SB-HN-145 | | | |
| *Mallotus apelta* Müll. Arg. | Wanning | SB-HN-135 | 18.69396 | 110.23170 | 118 |
| *Mallotus barbatus* Müll. Arg. | Sanya | SB-HN-136 | 18.26415 | 109.52084 | 119 |
| *Mallotus paniculatus* (Lam.) Müll. Arg. | Wanning | SB-HN-137 | 18.69396 | 110.23170 | 118 |
| *Melanolepis multiglandulosa* (Reinw. Ex Blume) Rchb. & Zoll. | Wanning | SB-HN-149 | 18.69396 | 110.23170 | 118 |
| *Ricinus communis* L. | Wanning | SB-HN-113 | 18.69396 | 110.23170 | 118 |
| **Subfamily Euphorbioideae** | | | | | |
| *Euphorbia heterophylla* L. | Chang Jiang | SB-HN-120 | 19.33905 | 108.210 | 33 |
| *Excoecaria cochinchinensis* Lour. | Sanya, Qiong Zhong | SB-HN-122 | 19.25343 | 109.38799 | 445 |
| *Hura crepitans* L. | Wanning | SB-HN-130 | 18.69396 | 110.23170 | 118 |
| *Triadica cochinchinensis* Lour. | Bawangling | SB-HN-140 | 19.112612 | 109.082207 | 650 |
| *Triadica sebifera* (L.) Small | Chang Jiang | SB-HN-142 | 19.33905 | 108.210 | 33 |
| **Subfamily Crotonoideae** | | | | | |
| *Aleurites moluccanus* Willd. | ChangJiang | SB-HN-10 | 19.33905 | 108.210 | 33 |
| *Codiaeum variegatum* (Linn.) Bl. (I) | Beihualing | SB-HN-16 | 19.00262 | 109.81523 | 471 |
| *Codiaeum variegatum* (Linn.) Bl. (II) | Wanning | SB-HN-23 | 18.69396 | 110.23170 | 90 |
| *Codiaeum variegatum* (Linn.) Bl. (III) | Sanya | SB-HN-121 | 18.27725 | 109.50888 | 12 |
| *Croton flavens* L. | ChangJiang | SB-HN-26 | 19.33905 | 108.210 | 33 |
| *Croton hirtus* L'Hér. | ChangJiang | SB-HN-27 | 19.33905 | 108.210 | 33 |
| *Endospermum chinense* Benth. | Haikou | SB-HN-29 | 20.062363 | 110.3186713 | 3 |
| *Hevea brasiliensis* (Willd. ex A. Juss.) Müll. Arg. | Wanning | SB-HN-30 | 18.69396 | 110.23170 | 90 |
| *Jatropha integerrima* Jacq. (I) | Haikou | SB-HN-9 | 20.062363 | 110.3186713 | 3 |
| *Jatropha integerrima* Jacq. (II) | Wanning | SB-HN-36 | 18.69396 | 110.23170 | 118 |
| *Manihot esculenta* Crants | Wanning | SB-HN-37 | 18.69396 | 110.23170 | 118 |
| *Suregada glomerulata* Baill. | Bawangling | SB-HN-141 | 19.112612 | 109.082207 | 650 |
| *Suregada multiflora* Baill. | Wanning | SB-HN-39 | 18.69396 | 110.23170 | 118 |
| *Vernicia montana* Lour. | Beihualing | SB-HN-17 | 19.00262 | 109.81523 | 471 |
| **Family Phyllanthaceae** | | | | | |
| *Aporosa dioca* (Roxb.) Müll. Arg. | Haikou | SB-HN-19 | 20.062363 | 110.3186713 | 3 |
| *Antidesma bunius* Wall. | Wanning | SB-HN-41 | 18.69396 | 110.23170 | 118 |
| *Antidesma montanum* Blume (I) | Wanning | SB-HN-49 | 18.69396 | 110.23170 | 90 |
| *Antidesma montanum* Blume (II) | Wanning | SB-HN-50 | 18.69396 | 110.23170 | 90 |
| *Baccaurea ramiflora* Lour. | Beihualing | SB-HN-51 | 19.00262 | 109.81523 | 471 |
| *Bischofia javanica* Blume (I) | Beihualing | SB-HN-53 | 19.00262 | 109.81523 | 471 |
| *Bischofia javanica* Blume (II) | Wanning | SB-HN-54 | 18.69396 | 110.23170 | 90 |
| *Breynia rostrata* Merr. | Wanning | SB-HN-57 | 18.69396 | 110.23170 | 90 |
| *Bridelia balansae* Tutcher | Wanning | SB-HN-58 | 18.69396 | 110.23170 | 90 |
| *Bridelia tomentosa* Blume | Wanning | SB-HN-60 | 18.69396 | 110.23170 | 90 |
| *Cleistanthus sumatranus* (Miq.) Müll. Arg. | Wanning | SB-HN-63 | 18.69396 | 110.23170 | 90 |
| *Glochidion sphaerogynum* (Müll. Arg.) Kurz | Wanning | SB-HN-64 | 18.69396 | 110.23170 | 90 |
| *Glochidion wrightii* Benth. | Wanning | SB-HN-65 | 18.69396 | 110.23170 | 90 |
| *Glochidion zeylanicum* (Gaertn.) A. Juss. | Bawangling | SB-HN-146 | 19.112612 | 109.082207 | 650 |

### 2.3. Light Microscopy (LM)

For the LM analysis, mature leaves of the Euphorbiaceae and Phyllanthaceae taxa were used. The species were investigated using previously published protocol [64]. About five mature leaf samples were selected for each species. To keep the specimens fresh, the mature leaf was placed in water for about 2 h. The leaf was cut into small pieces and put into a test tube containing a 30% nitric acid and 70% lactic acid solution. Later, the sample was boiled for about five minutes and was constantly shaken until the chlorophyll contents were removed. The transparent leaf sample was then poured into a Petri dish and washed with water first and then lactic acid to remove the debris from the epidermis. The adaxial and abaxial surface were separated using a camel brush. The epidermis was put on a glass slide and covered with a cover slip. The margin of the cover slip on the glass slide was coated with transparent nail polish to make the slides permanent. Finally, the

prepared slides were then observed under a light microscope (Xtl-10c computerized biological microscope; Shanghai Pudan optical instrument) with an 40× eyepiece. The LM micrographs were taken with the camera attached to the LM. Various morphological traits (nonglandular trichomes, different types of glandular trichomes, stomata type, ornamentation, and epidermal cells) were observed and incorporated.

## 2.4. Scanning Electron Microscopy (SEM)

For the SEM analysis, the middle portion of the mature leaf was selected. The sample was first washed with water and then ethanol to remove the debris by following the previously published protocol [64]. Both the upper and lower surface was air-dried and then put on a stub attached with double-sided sticky tape. Furthermore, the leaf samples were sputtered with Platinum for 25 min using a Leica Mikrosystem GmbH (Wien, Austria), made in Austria, and a high vacuum coater (ACE600) and observed under a scanning electron microscope (Thermo Scientific (Waltham, MA, USA), Model: verios g 4uc) installed in the analytical and testing center of the Hainan University Haikou China.

## 2.5. Statistical Analysis

For the statistical analysis, we used the leaf anatomical traits that include the presence and absence of nonglandular trichomes (NGTs), glandular trichomes (GTs) and stomata at both adaxial and abaxial surfaces. To observe the interrelationship between the foliar epidermal traits and species, a multivariate principal component analysis, a detrended correspondence analysis, a box plot and a chord diagram were generated by using both Originpro version 2021 and Past software.

## 3. Results

An attempt has been made to gather and explore the information of taxonomic potential concerned with the leaf epidermis of the family Euphorbiaceae and Phyllanthaceae. A total of 38 species were analyzed using both light microscopy (LM) and scanning electron microscopy (SEM). The scanning electron microscopic (SEM) micrographs are illustrated in Figures 1–11 whereas the light microscopic (LM) micrographs are illustrated in Figures 12 and 13. The quantitative data are presented in Tables 2–4 and illustrated in Figures 14–18.

### 3.1. Epidermis

Most of the Acalyphoideae taxa leaves are hypostomatic (i.e., the stomata are restricted to the lower surface in *Acalypha indica*, *A. hispida*, *A. wilkesiana*, *Ricinus communis*, *Mallotus apelta*, *M. barbatus* and *M. paniculatus*). However, the *Macaranga corymbosa* was found as amphistomatic. In the subfamily Euphorbioideae, *Excoecaria cochinchinensis*, *Hura crepitans*, *Triadica cochinchinensis*, and *Triadica sebifera* leaves were hypostomatic while the *Euphorbia heterophylla* leaf was found as amphistomatic. In the subfamily Crotonoideae, all twelve species have hypostomatic leaves. Similarly, in the family Phyllanthaceae, most of the species were hypostomatic except for *Bridellia tomentosa* which has amphistomatic leaves.

### 3.2. Nonglandular Trichomes (NGTs)

In the subfamily Acalyphoideae, nonglandular trichomes at both adaxial and abaxial surfaces were found in *Mallotus paniculatus*. At the abaxial surface, NGTs were found in *Acalypha indica*, *Mallotus barbatus*, *Mallotus paniculatus*, and *Macaranga corymbosa*. No NGTs were observed at the adaxial surface in all of the taxa except for *Mallotus paniculatus*. NGTs were absent in *Acalypha hispida*, *Acalypha wilkesiana*, *Ricinus communis*, and *Mallotus apelta*. In *Macaranga corymbosa*, a short conical-shaped NGT with a scabrate surface was infrequently distributed at the abaxial surface. In the species *Mallotus barbatus* and *M. paniculatus*, two types of NGTs were noted, the stellate and compactly arranged long thread-like NGTs. However, in *Mallotus paniculatus* the NGTs had pointed tips and stellate NGTs had 8–10 arms that delimited the *M. barbatus* which has 5–6 arms in stellate NGTs. Falcate-

shaped NGTs with a scabrate surface at the abaxial surface were infrequently distributed on midvein in *Acalypha indica* (Figures 1 and 2, Table 2).

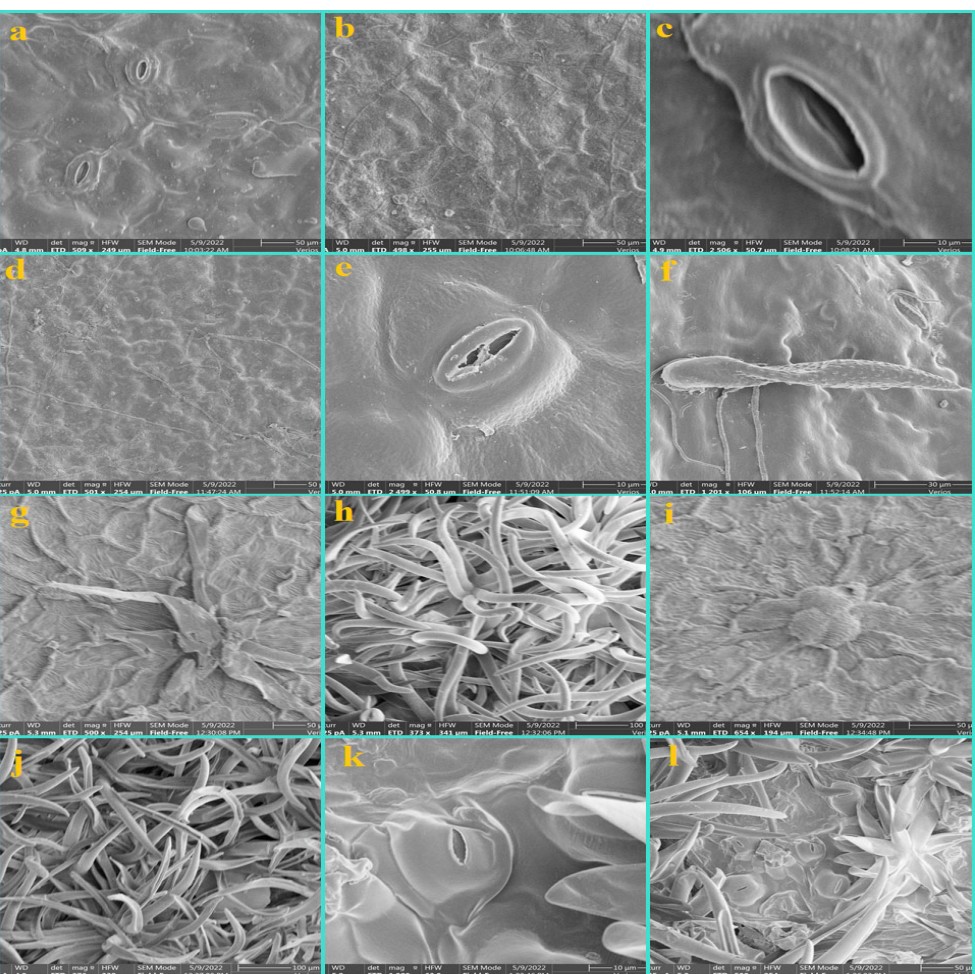

**Figure 1.** The SEM micrographs of the Acalyphoideae taxa. *Acalypha wilkesiana* (**a–c**), (**a**) general distribution of stomata, (**b**) epidermal cells, (**c**) a close view of stomata. *Alchornea rugosa* (**d–f**), (**d**) epidermal cells, (**e**) oriented stomata, (**f**) unicellular and nonglandular trichomes with a papillose surface. *Mallotus apelta* (**g,h**), (**g**) nonglandular trichomes, (**h**) stellate nonglandular trichomes intermingled with one another at the abaxial surface. *Mallotus paniculatus* (**i–l**), (**i**) GTs at the adaxial surface, (**j**) thick covering of NGTs at the abaxial surface (**k**), stomata with a smooth surface, (**l**) stellate NGTs at the abaxial surface.

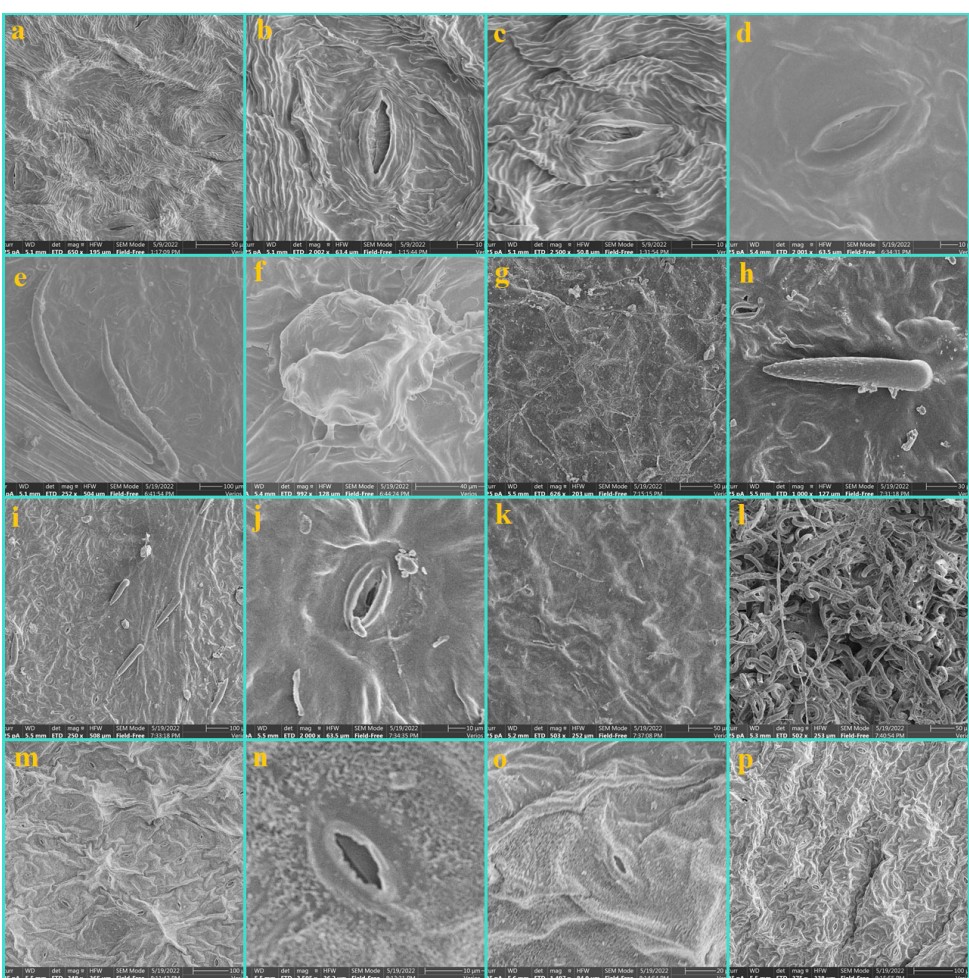

**Figure 2.** The SEM micrographs of the Acalyphoideae taxa. *Ricinus communis* (**a–c**), (**a**) general view of the epidermis showing cuticular striation at the abaxial surface, (**b**,**c**) stomata cuticular striations. *Acalypha indica* (**d–f**), (**d**) oriented stomata at the abaxial surface, (**e**) NGTs distributed on midvein at the abaxial surface, (**f**) subsessile capitate glandular trichomes. *Acalypha hispida* (**g**) epidermal cells at the adaxial surface. *Macaranga corymbosa* (**h–k**), (**h**) the unicellular conical-shape of NGTs at the abaxial surface, (**i**) general view of the abaxial surface in which the NGTs are distributed on midvein, (**j**) sunken stomata, (**k**) irregular shape epidermal cell at the adaxial surface. *Mallotus barbatus* (**l**), (**l**) a thick covering of NGTs at the abaxial surface. *Acalypha indica* (**m–p**), (**m**) a general view of the adaxial epidermis, (**n**,**o**) stomata with a papillose surface, (**p**) an abaxial surface.

**Table 2.** The presence (1) and absence (0) of the stomata and trichomes at both adaxial and abaxial surfaces.

| Species | Stomata | | NGTS | | GTs | |
|---|---|---|---|---|---|---|
| | Adaxial | Abaxial | Adaxial | Abaxial | Adaxial | Abaxial |
| **Acalyphoideae** | | | | | | |
| *Acalypha hispida* | 0 | 1 | 0 | 0 | 0 | 0 |
| *Acalypha indica* | 0 | 1 | 0 | 1 | 0 | 1 |
| *Acalypha wilkesiana* | 0 | 1 | 0 | 0 | 0 | 0 |
| *Alchornea rugosa* | 0 | 1 | 0 | 1 | 0 | 1 |
| *Macaranga corymbosa* | 1 | 1 | 0 | 1 | 0 | 1 |
| *Mallotus apelta* | 0 | 1 | 0 | 0 | 0 | 0 |
| *Mallotus barbatus* | 0 | 1 | 0 | 1 | 1 | 1 |
| *Mallotus paniculatus* | 0 | 1 | 1 | 1 | 0 | 0 |
| *Ricinus communis* | 0 | 1 | 0 | 0 | 0 | 0 |

| | | | | | | |
|---|---|---|---|---|---|---|
| **Euphorbioideae** | | | | | | |
| *Euphorbia heterophylla* | 1 | 1 | 0 | 1 | 1 | 1 |
| *Excoecaria cochinchinensis* | 0 | 1 | 0 | 0 | 1 | 1 |
| *Hura crepitans* | 0 | 1 | 0 | 0 | 0 | 0 |
| *Triadica cochinchinensis* | 0 | 1 | 0 | 0 | 1 | 0 |
| *Triadica sebifera* | 0 | 1 | 0 | 0 | 0 | 0 |
| **Crotonoideae** | | | | | | |
| *Aleurites moluccana* | 0 | 1 | 0 | 0 | 1 | 0 |
| *Codiaeum variegatum* (I) | 0 | 1 | 0 | 0 | 0 | 1 |
| *Codiaeum variegatum* (II) | 0 | 1 | 0 | 0 | 0 | 0 |
| *Codiaeum variegatum* (III) | 0 | 1 | 1 | 0 | 0 | 0 |
| *Croton flavens* | 0 | *0* | 0 | 1 | *0* | *0* |
| *Croton hirtus* | 1 | 1 | 0 | 1 | 0 | 1 |
| *Endospermum chinense* | 0 | 1 | 0 | 0 | 1 | 0 |
| *Hevea brasiliensis* | 0 | 1 | 0 | 0 | 0 | 0 |
| *Jatropha integerrima* (I) | 0 | 1 | 0 | 0 | 0 | 0 |
| *Jatropha integerrima* (II) | 0 | 1 | 0 | 0 | 0 | 0 |
| *Manihot esculenta* | 0 | 1 | 0 | 0 | 1 | 1 |
| *Suregada glomerulata* | 0 | 1 | 0 | 0 | 0 | 0 |
| *Suregada multiflora* | 0 | 1 | 0 | 1 | 0 | 1 |
| *Vernicia montana* | *0* | 1 | *0* | *0* | *0* | *1* |
| **Phyllanthaceae** | | | | | | |
| *Antidesma bunius* | 0 | 1 | 0 | 0 | 1 | 0 |
| *Antidesma montanum* (I) | 0 | 1 | 0 | 1 | 0 | 1 |
| *Antidesma montanum* (II) | 0 | 1 | 0 | 0 | 0 | 0 |
| *Aporosa dioca* | 0 | 1 | 0 | 1 | 1 | 0 |
| *Baccaurea ramiflora* | 0 | 1 | 0 | 0 | 0 | 0 |
| *Bischofia javanica* | 0 | 1 | 0 | 1 | 0 | 0 |
| *Bischofia javanica* | 0 | 1 | 0 | 0 | 0 | 0 |
| *Breynia rostrata* | 0 | 1 | 0 | 1 | 0 | 0 |
| *Bridelia balansae* | 0 | 1 | 1 | 1 | 0 | 1 |
| *Bridelia tomentosa* | 1 | 1 | 1 | 1 | 0 | 0 |
| *Cleistanthus sumatranus* | 0 | 1 | 0 | 0 | 0 | 0 |
| *Glochidion sphaerogynum* | 0 | 1 | 0 | 0 | 0 | 0 |
| *Glochidion wrightii* | 0 | 1 | 0 | 0 | 0 | 0 |
| *Glochidion zeylanicum* | 0 | 1 | 0 | 1 | 0 | 0 |

**NGTs;** Nonglandular trichomes, **GTs;** Glandular trichomes.

Similarly, in the subfamily Crotonoideae, the NGTs were absent in most of the species. At the adaxial surface, long multicellular NGTs were occasionally seen in *Codiaeum variegatum*, while at the abaxial surface, it was seen in *Croton hirtus* and *Suregada multiflora*. At both adaxial and abaxial surfaces, the NGTs were found in *Croton flavens*. In *Croton hirtus*, highly branched stellate NGTs with arms ranging 8–13 were observed with rich density at the abaxial surface. The small papilose NGTs were occasionally found in *Croton flavens* at the adaxial surface. The lepidote-subentire NGTs with an oriented center were found in *Croton flavens* and covered the entire abaxial surface while infrequently distributed at the adaxial surface and should be considered as a potential taxonomic trait for the classification of the species. The long falcate-shaped NGTs with scabrate surfaces were infrequently distributed on the midvein on the abaxial surface in *Suregada multiflora* (Figures 3–6).

In the subfamily Euphorbioideae, the multicellular 2–4 celled long conical-shaped NGTs were infrequently distributed on the abaxial surface in *Euphorbia heterophylla* while absent in the rest of the species. While *Excoecaria cochinchinensis*, *Hura crepitans*, *Triadica sebifera* and *T. cochinchinensis* were absent (Figure 7).

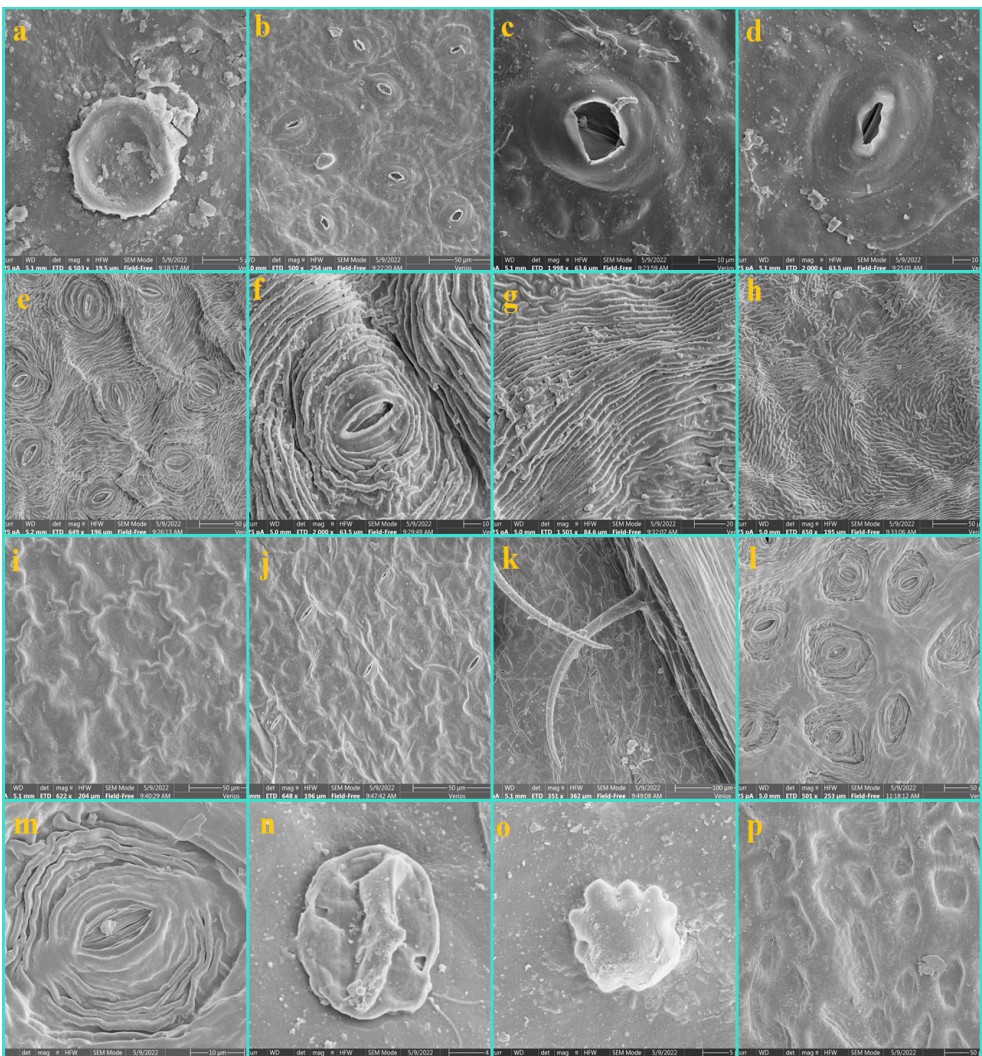

**Figure 3.** The SEM micrographs of the foliar epidermis of Crotonoideae taxa. *Codiaeum variegatum* (**a–d**), (**a**) sunken GTs at the adaxial surface, (**b**) the distribution of stomata at the abaxial surface, (**c,d**) the orientation of the stomata and stomatal pore wideness. *Jatropha integerrima* (**e–h**), (**e**) the distribution of stomata at the abaxial surface, (**f**) cuticular striation parallel to the length of stomata, (**g,h**) cuticular striation at the adaxial surface. *Suregada multiflora* (**i–k**), (**i**) irregular epidermal cells at the adaxial surface, (**j**) the distribution of stomata at the abaxial surface, (**k**) falcate NGTs on midvein at the abaxial surface. *Endospermum chinense* (**l–p**), (**l**) sunken stomata at the abaxial surface, (**m**) close view of the stomata with cuticular striations, (**n**) sessile capitate GTs at the adaxial surface, (**o**) subsessile GTs at the adaxial surface, (**p**) epidermal cells at the adaxial surface.

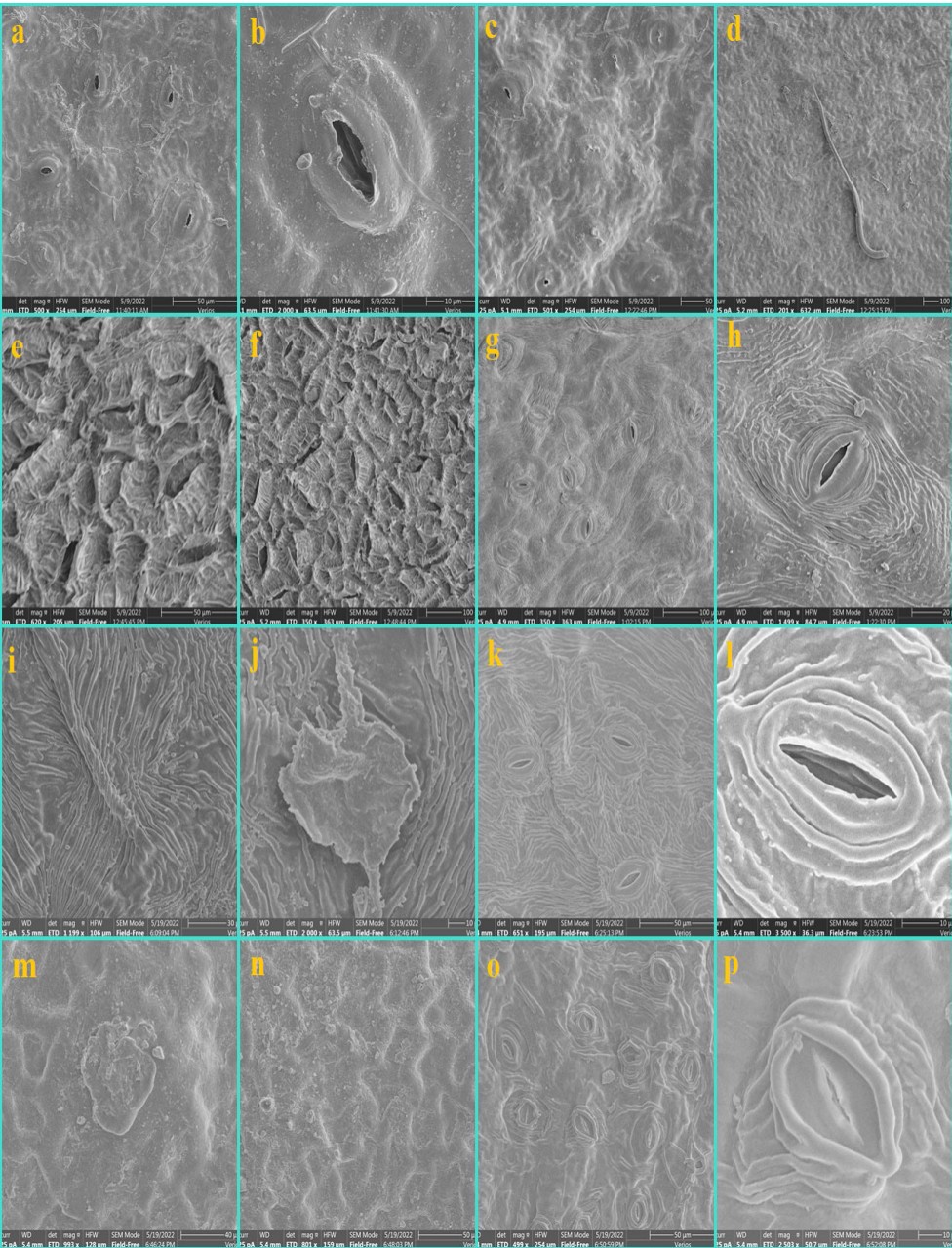

**Figure 4.** The SEM micrographs of the foliar epidermis of Crotonoideae taxa. *Codiaeum variegatum* (I) (**a**,**b**), (**a**) stomata at the abaxial surface, (**b**) close view of stomata. *Codiaeum variegatum* (II) (**c**,**d**), (**c**) general view of the abaxial surface, (**d**) adaxial surface. *Hevea brasiliensis* (**e**,**f**), (**e**) *Jatropha integerrima* (I), *Jatropha integerrima* (II) (**g**,**h**), *Endospermum chinense* (**i**–**l**), *Aleurites moluccana* (**m**–**p**). (**m**,**n**) general view of the adaxial surface, (**o**) distribution of stomata at the adaxial surface, (**p**) wide stomatal opening.

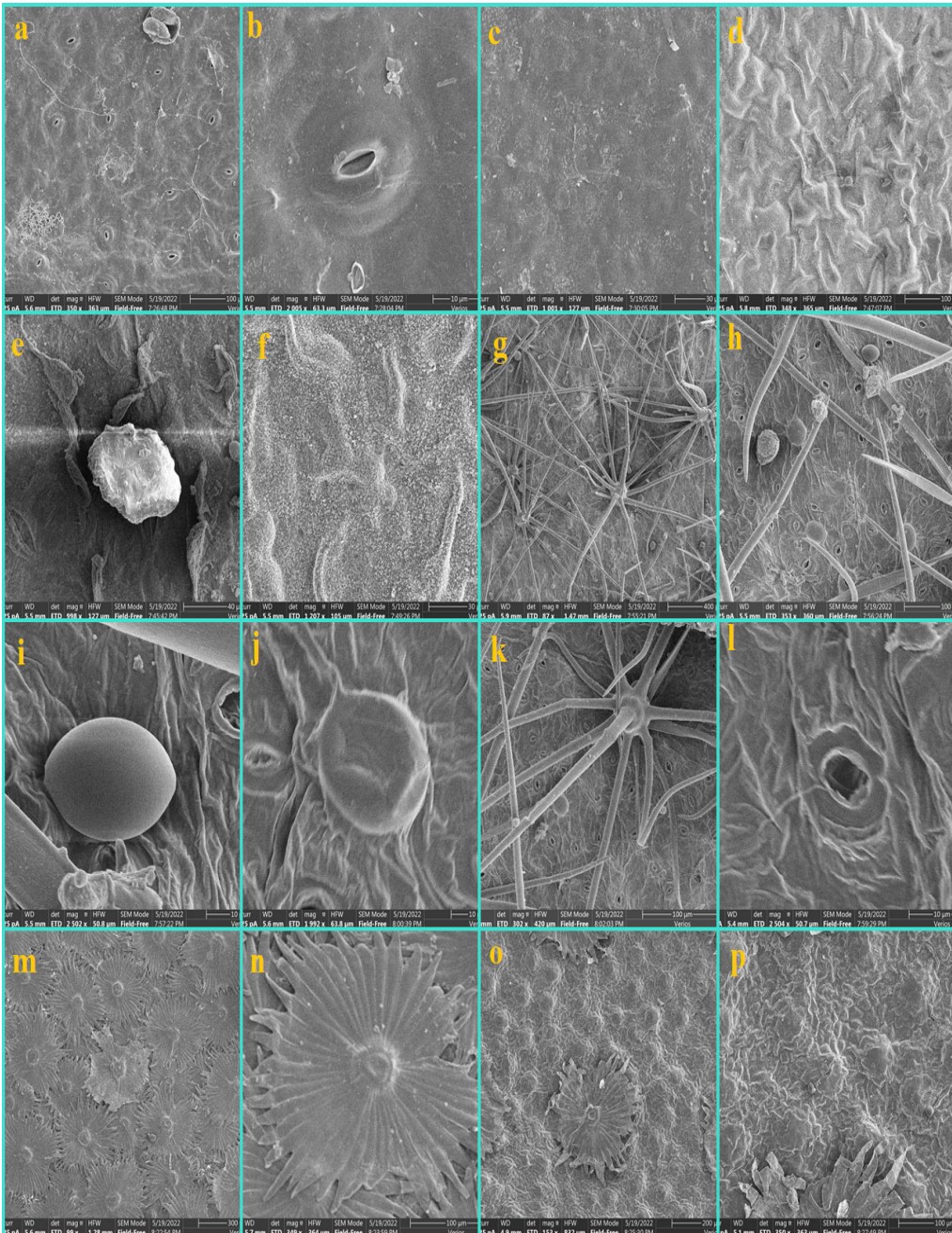

**Figure 5.** The SEM micrographs of the foliar epidermis of Crotonoideae taxa. *Suregada glomerulata* (**a**–**c**), (**a**) distribution of stomata at the abaxial surface, (**b**) sunken and close view of stomata, (**c**) the adaxial surface. *Manihot esculenta* (**d**–**f**), (**d**) adaxial surface showing no stomata (**e**) flat NGTs at the abaxial (**f**) papillose adaxial surface. *Croton hirtus* (**g**–**l**), (**g**,**h**) stellate NGTs at the abaxial surface, (**i**,**j**) subsessile GTs (**k**) stellate NGTs, (**l**) close view of stomata. *Croton flavens* (**m**–**p**). (**m**) thick covering of lepidote trichomes at the abaxial surface, (**n**) close view of the lepidote trichomes, (**o**,**p**) adaxial surface.

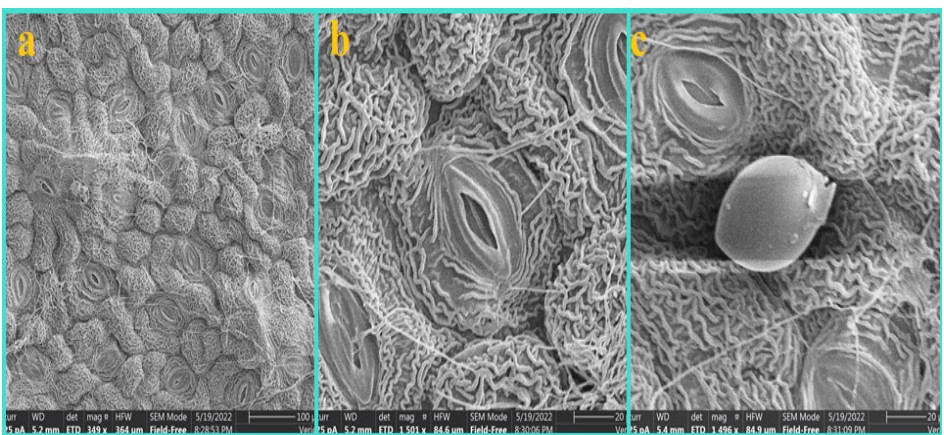

**Figure 6.** The SEM micrographs of the foliar epidermis of Crotonoideae taxa. *Vernicia montana* (**a**–**c**), (**a**) general view of the abaxial surface, (**b**) close view of stomata with reticulate cuticular striation, (**c**) subsessile GTs.

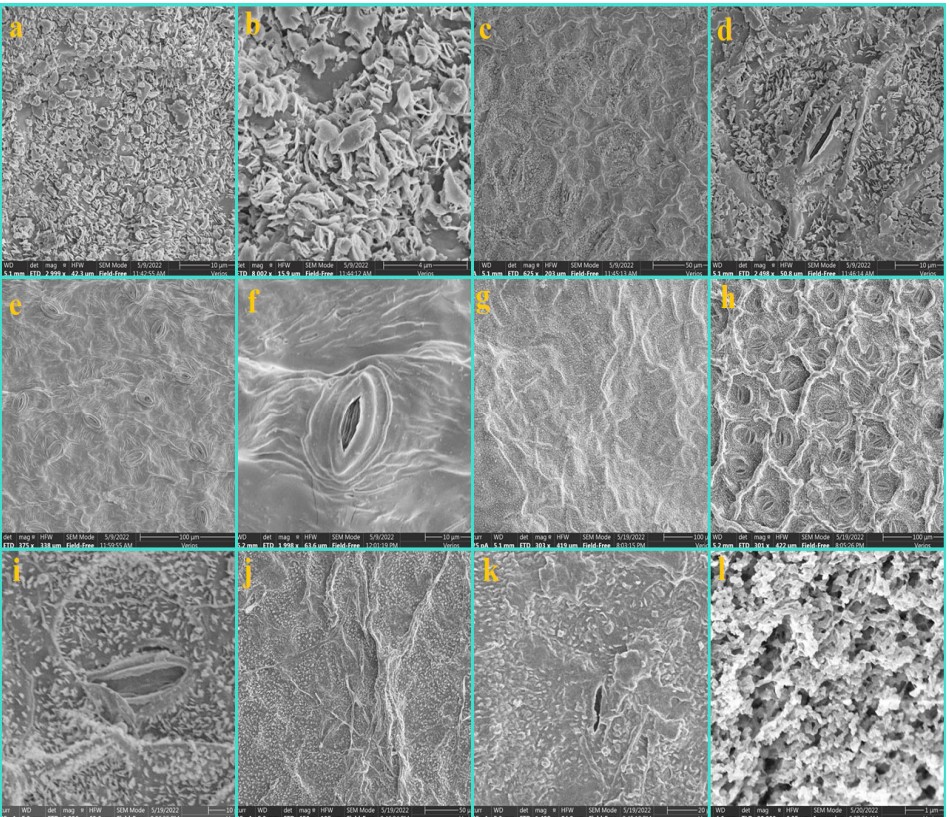

**Figure 7.** The SEM micrographs of the foliar epidermis of Euphorbioideae taxa. *Triadica cochinchinensis* (sample 1) (**a**–**d**), (**a**,**b**) the adaxial rough surface, (**c**) distribution of stomata at the abaxial surface, (**d**) close view of stomata with papillose surface *Hura crepitans* (**e**,**f**), abaxial surface, (**f**) close view of stomata. *Triadica sebifera* (**g**–**i**), (**g**) adaxial surface, (**h**) distribution of stomata at the abaxial surface, (**i**) sunken stomata with a papillose surface. *Triadica cochinchinensis* (sample 2) (**j**–**l**), (**j**) adaxial surface, (**k**) presence of stomata, (**l**) abaxial surface.

In the family Phyllanthaceae, the NGTs were found in most of the species. At both adaxial and abaxial surfaces, they were found in *Bridelia balansae* and *B. tomentosa*. Similarly, at the abaxial surface, the NGTs were noted in *Antidesma montanum* (II), *Bischofia javanica*, *Bridelia balansae*, *B. tomentosa*, *Glochidion zeylanicum*, *Aporosa dioca*, and *Breynia rostrata*. The NGTs were

absent in *Baccaurea ramiflora*, *Bischofia javanica*, *Glochidion wrightii*, *G. sphaerogynum*, and *Cleistanthus sumatranus*. The short and long unicellular nonglandular trichomes (NGTs) at the abaxial surface on the midvein were found in *Aporosa dioca*. The multicellular, long and papilose-type NGTs were found in *Breynia rostrata* at the abaxial surface with thick density. Long and multicellular NGTs with scabrate surface were rarely found at the abaxial surface in *Antidesma montanum* (I). Short conical shaped in *Bischofia javanica* and *Alchornea rugosa*, long falcate shaped and multicellular NGTs were noted on the midvein at the abaxial surface in *Aporosa dioca* and the adaxial surface in *Bridelia balansae*. However, the multicellular and flattened NGTs at the abaxial surface were seen in *Bridelia balansae*. While in *Bridellia tomentosa* the long falcate-shaped NGTs were infrequently found at the adaxial and densely distributed mostly on a network of the veins at the abaxial surface sometimes intermingled with one another. Hence these features can easily differentiate the *Bridellia* species from one another. Short conical-shaped NGTs with a striate surface were rarely seen in *Glochidion zeylanicum* (Figures 8–11).

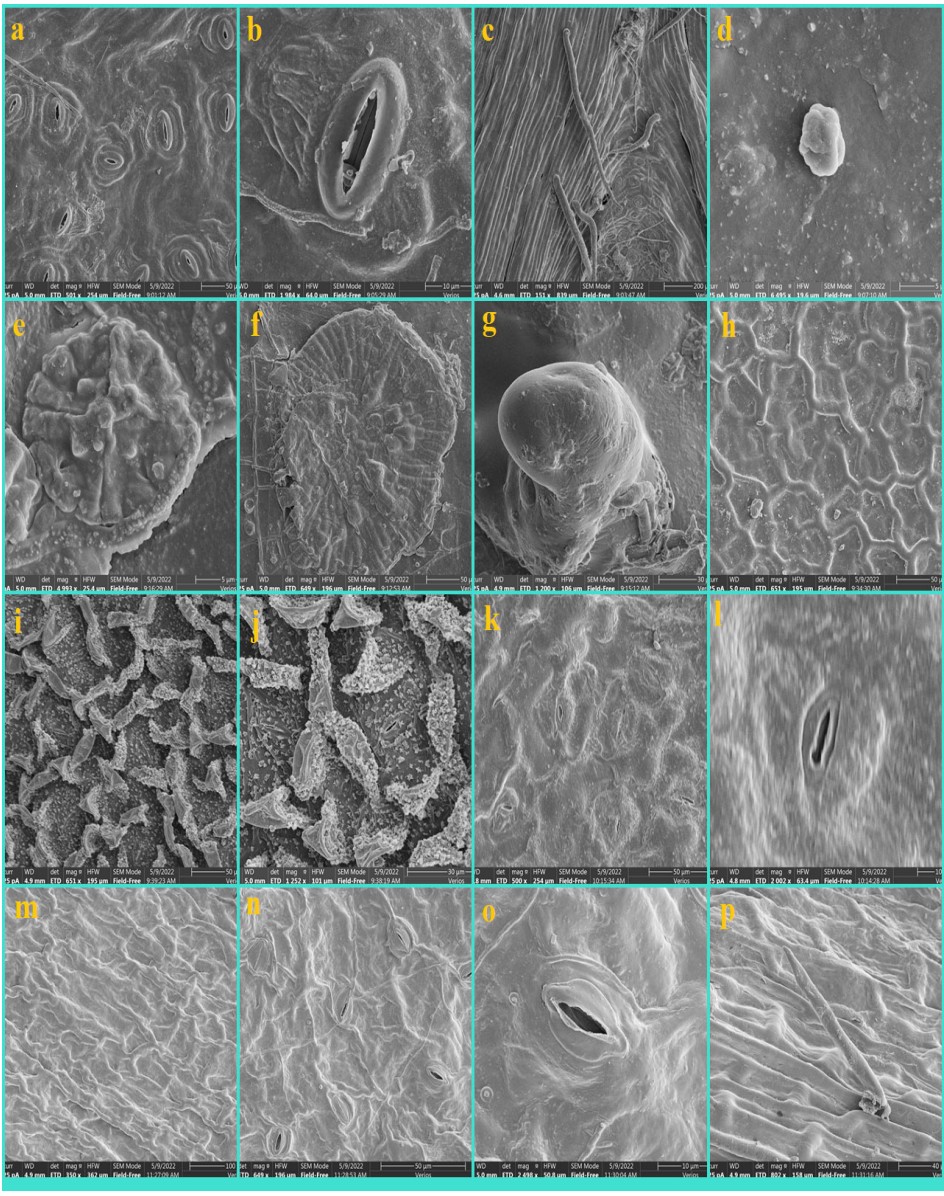

**Figure 8.** The SEM micrographs of the foliar epidermis of Phyllanthaceae taxa. *Aporosa dioca* (**a**–**g**), (**a**) adaxial surface, (**b**) close view of stomata, (**c**) distribution of NGTs at the abaxial surface, (**d**)

subsessile GTs at the adaxial surface, (**e**,**f**) peltate trichomes, (**g**) capitate GTs. *Breynia rostrata* (**h–j**), (**h**) epidermal cells at the adaxial surface, (**i**,**j**) flat NGTs with a scabrate surface. *Cleistanthus sumatranus* (**k**,**l**), (**k**) distribution of stomata at the abaxial surface, (**l**) sunken stomata. *Bischofia javanica* (sample 1) (**m–p**), (**m**) adaxial surface, (**n**) abaxial surface, (**o**) oriented stomata at the abaxial surface, (**p**) unicellular NGTs.

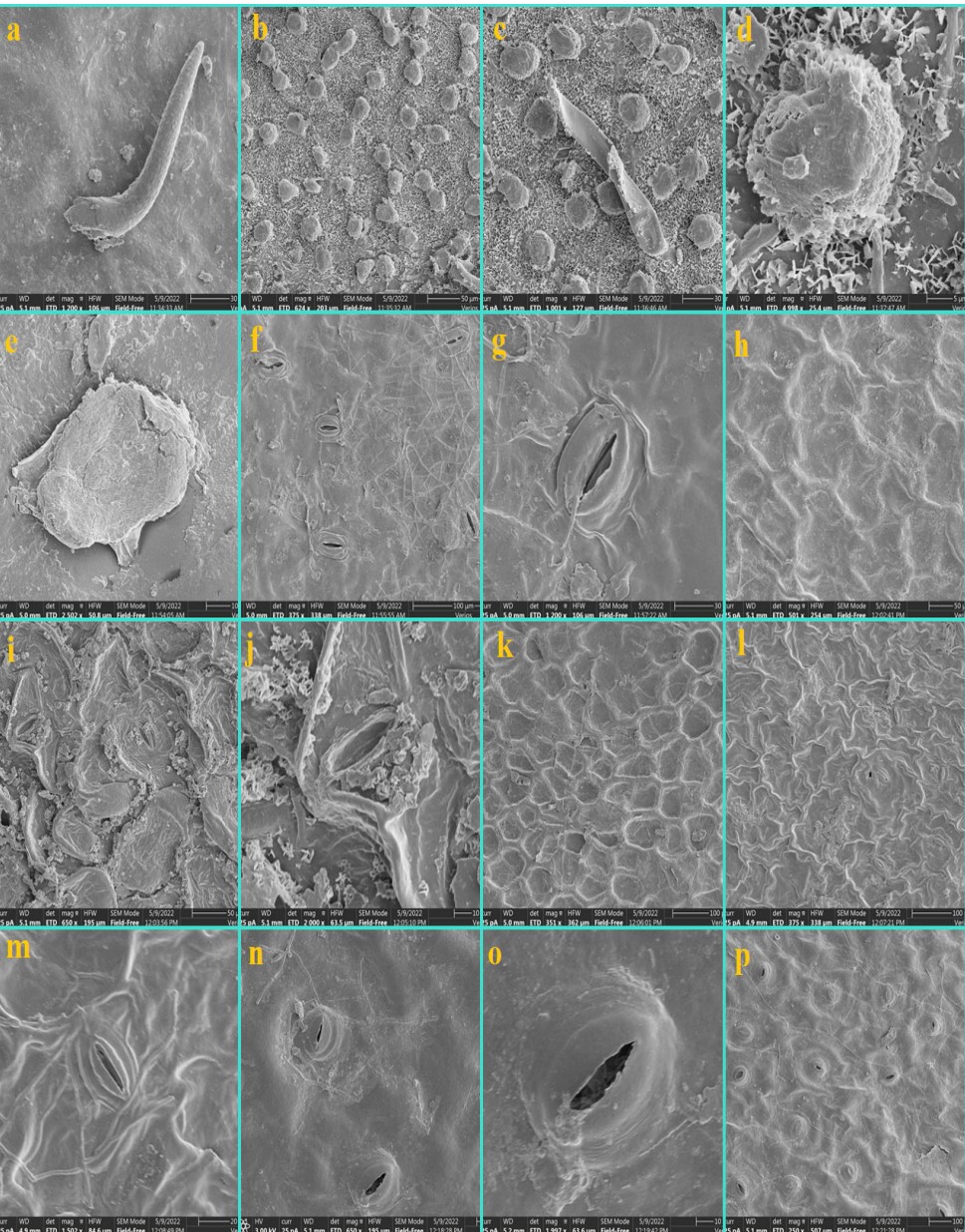

**Figure 9.** The SEM micrographs of the foliar epidermis of Phyllanthaceae taxa. *Bridelia balansae* (**a–d**), (**a**) unicellular conical-shaped NGTs, (**b**) thick distribution of GTs at the abaxial surface, (**c**) flat unicellular NGTs at the abaxial surface, (**d**) close view of subsessile GTs. *Antidesma bunius* (**e–g**), (**e**) capitate GTs at the adaxial surface, (**f**) distribution of stomata at the abaxial surface (**g**) close view of sunken stomata. *Glochidion wrightii* (**h–j**), (**h**) epidermal cells at the adaxial surface, (**i**) distribution of stomata at the abaxial surface, (**j**) close of stomata, *Glochidion sphaerogynum* (**k–m**), (**k**) polygonal-shaped epidermal cells at the adaxial surface, (**l**) irregular-shaped epidermal cells, (**m**) close view of oriented stomata, *Antidesma montanum* (sample 1) (**n–p**), (**n**) distribution of sunken stomata at the abaxial surface, (**o**) close view of stomata with a smooth surface, (**p**) the abaxial surface.

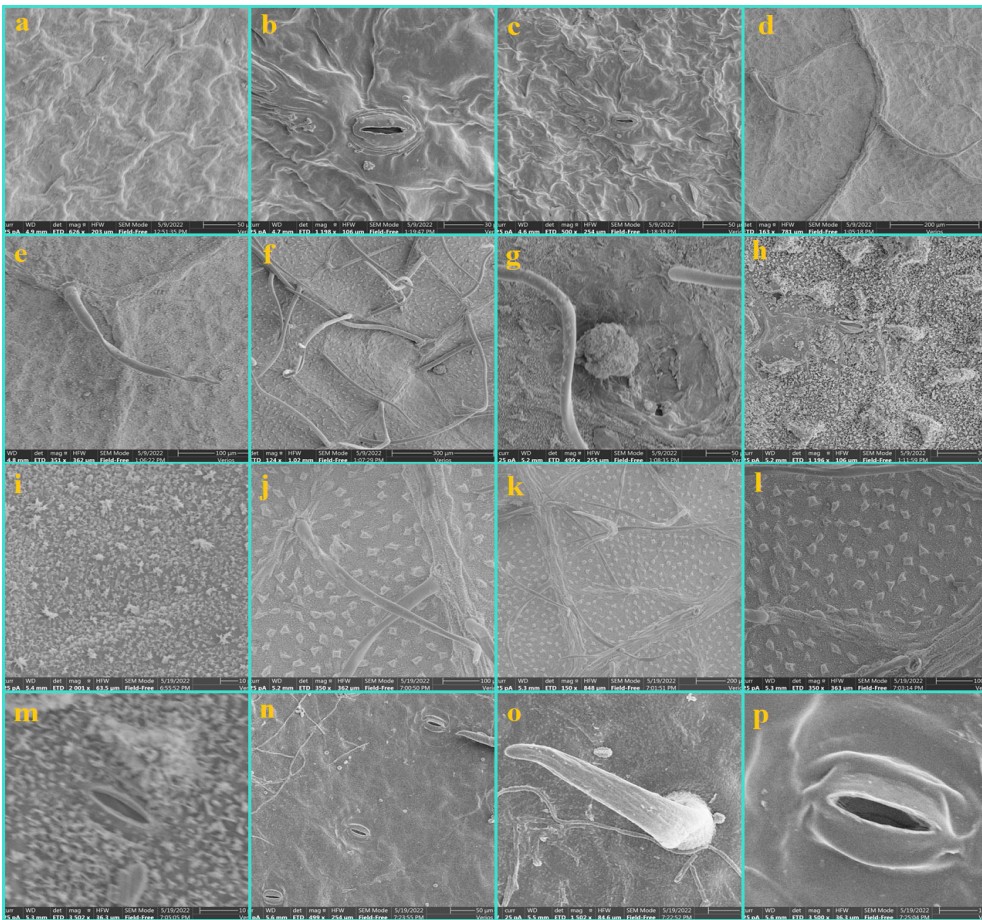

**Figure 10.** The SEM micrographs of the foliar epidermis of Phyllanthaceae taxa. *Bischofia javanica* (sample 2) (**a–c**), (**a**) the abaxial surface, (**b,c**) a general view of stomata at the adaxial surface. *Bridellia tomentosa* (sample 1), (**d–h**), (**d,e**) falcate-shaped NGTs on the vein at the adaxial surface, (**f**) NGTs mainly distributed on the veins at the abaxial surface (**g**) subsessile capitate GTs, (**h**) papillose epidermal cells at the abaxial surface with ridges. *Bridellia tomentosa* (sample 2), (**i–m**), (**i**) the adaxial surface with a papillose surface, (**j,k**) long falcate-shaped NGTs distributed on the main and secondary vein, (**l**) a general view of the abaxial surface with ridges, (**m**) sunken stomata with papillose ornamentation. *Glochidion zeylanicum* (**n–p**), (**n**) the abaxial surface, (**o**) unicellular NGTs, (**p**) a close view of rectangular-shaped stomata.

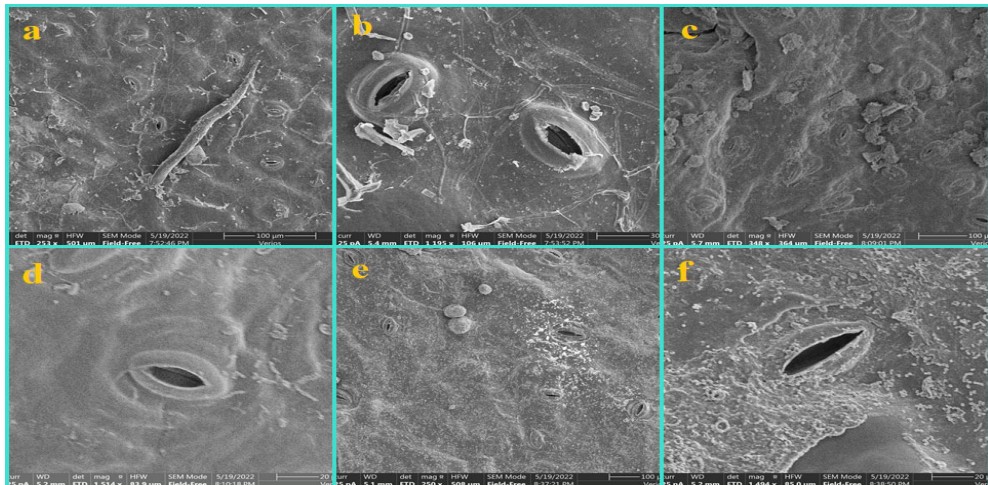

**Figure 11.** The SEM micrographs of the foliar epidermis of Phyllanthaceae taxa. *Antidesma montanum* (sample 2) (**a**,**b**), (**a**) stomata distribution at the abaxial surface, (**b**) close view of the stomata with a smooth surface at the abaxial surface, *Aporosa dioca* (**c**,**d**), (**c**) stomata distribution at the abaxial surface, (**d**) close view of stomata. *Antidesma bunius* (**e**,**f**), (**e**) general view of the abaxial surface, (**f**) sunken stomata.

### 3.3. Glandular Trichomes (GTs)

In the subfamily Acalyphoideae, the GTs were found at the abaxial surface in *Acalypha indica*, *Mallotus barbatus*, and *Macaranga corymbosa*. At both the abaxial and adaxial surface they was only observed in *Mallotus barbatus*. No GTs were found in *Acalypha hispida*, *A. wilkesiana*, *Ricinus communis*, *Mallotus apelta*, and *M. paniculatus* (Table 2). The subsessile capitate GTs with the unicellular head were found in *Acalypha indica* and *Mallotus barbatus* at the abaxial surface. In *Macaranga corymbosa* the capitate GTs with the multicellular head were observed at the abaxial surface (Figures 1 and 2).

In the subfamily Euphorboideae, the subsessile capitate GTs were only found in *Triadica cochinchinensis* at the adaxial surface. Both capitate and subsessile capitates GTs were found in *Euphorbia heterophylla*, however, they were absent in *Excoecaria cochinchinensis*, *Hura crepitans*, and *Triadica sebifera* (Figure 7).

In the subfamily Crotonoideae, the GTs were found at the adaxial surface in *Aleurites moluccana*, *Endospermum chinense*, and *Croton flavens*. Similarly, at the abaxial surface they were found in *Codiaeum variegatum*, *Vernicia montana*, *Croton hirtus*, *Suregada multiflora*, and *Manihot esculenta*. In *Aleurites moluccana*, the subsessile capitate GTs were occasionally observed at the adaxial surface. The sessile capitate GTs with multicellular heads were rarely found in *Codiaeum variegatum* at the abaxial surface and in *Endospermum chinense* at the adaxial surface. However, the subentire capitate GTs were additionally found in the latter species. Similarly, the subsessile capitate GTs with unicellular head were occasionally found at the abaxial surface in *Vernicia montana*. In *Croton hirtus* both capitate and subsessile capitate GTs with unicellular to multicellular heads were densely distributed at the abaxial surface. The capitate GTs with multicellular heads were rarely found in *Suregada multiflora* at the abaxial surface. While in *Manihot esculenta* a small-size subsessile capitate GTs were noted at the abaxial surface (Figures 3–6).

Similarly in the family Phyllanthaceae, the GTs were observed at the adaxial surface in *Antidesma bunius*, *Bridelia balansae*, and *Aporosa dioca*. Similarly, at the abaxial surface they was found in *Alchornea rugosa*, *Antidesma montanum*, and *Bridelia balansae* while no GTs were found in the rest of the species. Two types of GTs were found in *Aporosa dioca*, multicellular peltate and capitate glandular trichome at the adaxial surface. A capitate with a multicellular head and a short stalk was noted in *Antidesma bunius* at the abaxial surface, while subsessile capitate GTs with unicellular heads in *Antidesma montanum* (I) were taxonomically significant to define species boundaries within the genus. Similarly,

in *Bridelia balansae*, both subsessile and sessile capitate GTs were densely distributed at the abaxial surface, while sessile GTs were observed in *Alchornea rugosa* (Figures 8–12).

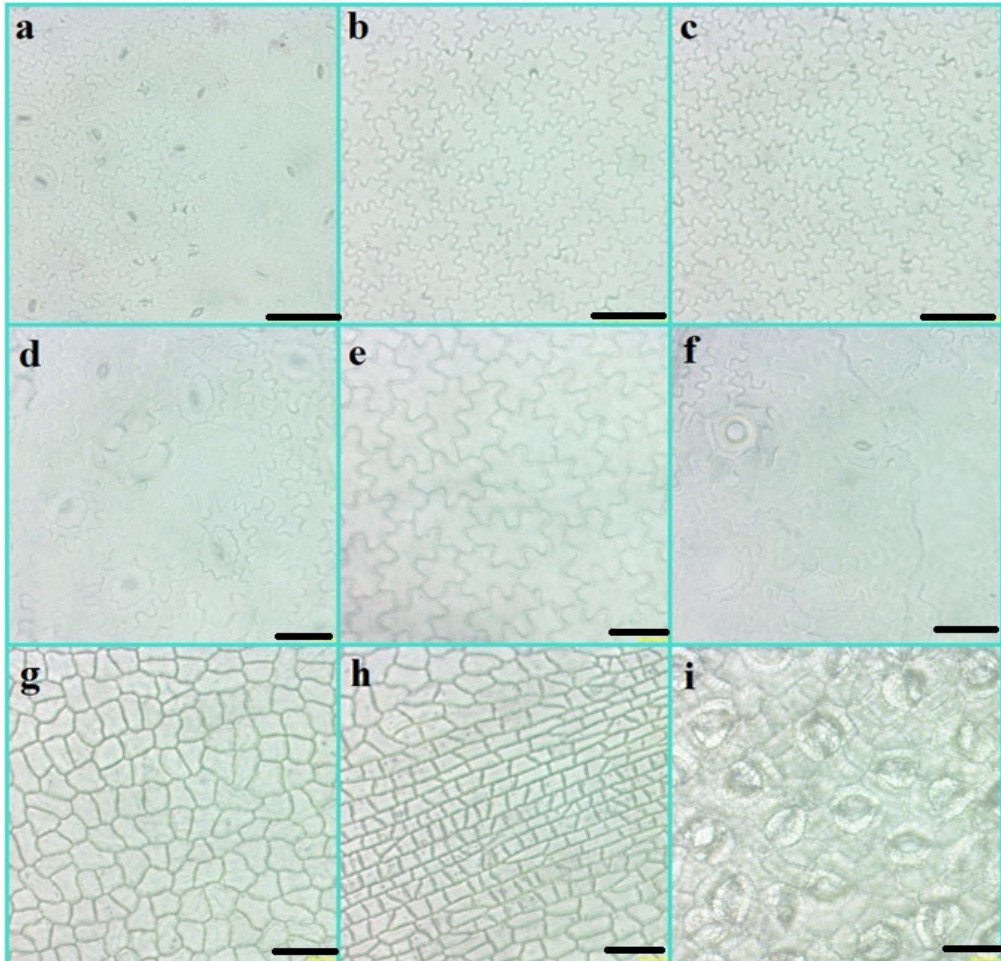

**Figure 12.** The LM micrographs of *Excoecaria cochinchinensis* (**a**–**c**), (**a**) abaxial surface, (**b**,**c**) deeply undulate epidermal cells at the adaxial surface. *Glochidion wrightii* (**d**–**f**), (**d**) the abaxial surface, (**e**) the adaxial surface, (**f**) the abaxial surface. *Hevea brasiliensis* (**g**–**i**), (**g**,**h**) the rectangular and polygonal epidermal cells at the adaxial surface, (**i**) the distribution of stomata at the abaxial surface. Yellow scale bars at the right bottom are 20 μm.

### 3.4. Shape and Anticlinal Wall Pattern of Epidermal Cells

The adaxial surface was covered by a thick cuticle with an anticlinal wall pattern that was straight to the sinus in *Acalypha hispida*, sinus in *Acalypha indica*, deeply undulate in *Mallotus apelta*, undulate in *Alchornea rugosa*, and straight to undulate in *Mallotus paniculatus* of the subfamily Acalyphoideae. The shape of the epidermal cells was irregular in most of the species and infrequently polygonal in *Acalypha hispida* and *A. wilkesiana*, and elongated in *Ricinus communis*.

In the subfamily Euphorbioideae the irregular-shaped epidermal cells were found in *Euphorbia heterophylla*, *Excoecaria cochinchinensis*, *Hura crepitans*, *Triadica cochinchinensis* and irregular to polygonal in *T. sebifera*. The sinus anticlinal wall pattern was found in *Euphorbia heterophylla* and *Triadica cochinchinensis*, straight in *Triadica sebifera*, deeply undulate in *Excoecaria cochinchinensis*, and undulate in *Hura crepitans*.

In Crotonoideae, the polygonal-shaped epidermal cells were observed in *Aleurites moluccana*, *Endospermum chinense*, in both samples of *Jatropha integerrima* and *Suregada glomerulata*. Polygonal to irregular shapes were observed in *Aleurites moluccana* and

*Endospermum chinense,* while the anticlinal wall patterns were straight to sinus in *Aleurites moluccana* and *Endospermum chinense,* slightly wavy in *Croton hirtus,* deeply undulate in *Codiaeum variegatum* (I) and *Manihot esculenta,* undulate in *Codiaeum variegatum* (II), Codiaeum variegatum (III), Croton flavens, and Suregada multiflora, sinus in Vernicia montana and Hevea brasiliensis, and straight in both samples of the Jatropha integerrima.

In the family Phyllanthaceae, the shape of the epidermal cell was polygonal to irregular in *Breynia rostrata* and *Glochidion sphaerogynum.* Polygonal in *Bridelia balansae* and irregular in the rest of the species. Similarly, the anticlinal wall pattern was undulate in *Antidesma bunius, A. montanum* (II), *Bischofia javanica* (II), *Glochidion wrightii, G. zeylanicum,* and *Cleistanthus sumatranus,* deeply undulate in *Baccaurea ramiflora,* undulate to sinus in *Bischofia javanica* (I), straight to undulate in *Glochidion sphaerogynum,* straight to sinus in *Breynia rostrata,* and straight in *Bridelia balansae* (Figure 13, Table 2).

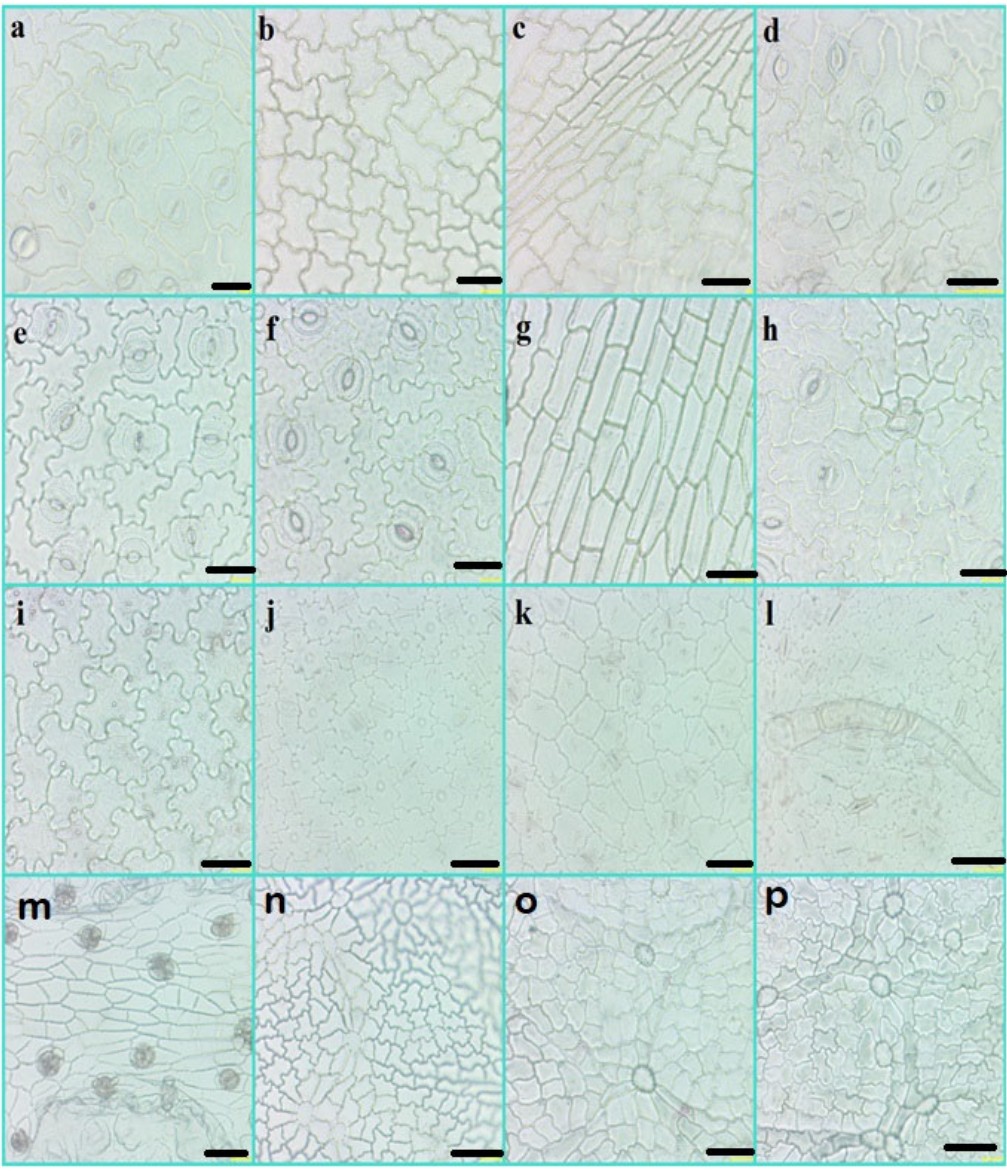

**Figure 13.** The LM micrographs of *Cleistanthus sumatranus* (**a–d**), (**a**) abaxial surface showing paracytic and anisocytic stomata, (**b,c**) epidermal cells at the adaxial surface, (**d**) adaxial surface. *Codiaeum variegatum* (sample 1) (**e–i**), (**e,f**) anomocytic stomata at the abaxial surface, (**g**) elongated epidermal cells at the adaxial surface, (**h**) GTs at the abaxial surface, (**i**) undulate epidermal cells at the abaxial surface. *Euphorbia heterophylla* (**j–l**), (**j**) the abaxial surface, (**k**) epidermal cells at the

adaxial surface, (**l**) four-cell long NGTs at the abaxial surface. *Mallotus barbatus* (**m–p**), (**m**) Elongated epidermal cells with GTs at the abaxial surface, (**n–p**) irregular and polygonal epidermal cells at the adaxial surface. Yellow scale bars at the right bottom are 20 μm.

Overall, the adaxial epidermis is overlaid by a thick cuticle layer. The number of epidermal cells and anticlinal wall sinuations was less, whereas the size of the epidermal cells was mostly large when compared to the abaxial surface. In contrast, for the abaxial surface the size of the epidermal cells were usually smaller with a comparatively thinner cuticle with more pronounced anticlinal wall sinuations than the adaxial surface.

### 3.5. Types of Stomata

In the analyzed taxa of Acalyphoideae, variation was found in the type and orientation of the stomata. For example, the arrested and anomocytic mixed with anisocytic stomata were found in *Acalypha hispida*, and anomocytic in *A. indica* and *Macaranga corymbosa*. A small size and sunken stomata with a smooth surface were noted for *Macaranga corymbosa*. The anisocytic-type stomata were noted in *Acalypha wilkesiana*, *Ricinus communis*, and *Mallotus barbatus*. Small size and sunken stomata were found in *Macaranga corymbosa* with a psilate surface. However, large-sized and oriented stomata with cuticular striae extending as lateral wings, relatively wide and elliptical-shaped stomatal pores, and conspicuous outer stomatal ledges were found in *Ricinus communis*. Small size and oriented paracytic stomata were found in *Mallotus paniculatus*. However, the stomata were not visible at the abaxial surface in *Mallotus apelta* due to a thick covering of NGTs intermingled with one another that protected the epidermis.

In the subfamily Euphorbioideae, the anomocytic and anisocytic stomata were found in *Euphorbia heterophylla*, anisocytic in *Excoecaria cochinchinensis*, and anomocytic in *Hura crepitans*, *Triadica cochinchinensis*, and *T. sebifera*. The large-sized and oriented stomata in *Hura crepitans* were different with a smooth surface and cuticular striation radiating from the guard cells. However, sunken and large-sized stomata with rich density and densely papilose surfaces were found in *Triadica sebifera*, while the same features were noted for *T. cochinchinensis* except they had less density at the abaxial surface.

In the subfamily Crotonoideae, the paracytic type of stomata was found in most of the species. In *Croton hirtus* the aborted and sunken anisocytic and anisocytic were observed. It was anomocytic in *Hevea brasiliensis*, anomocytic mixed with paracytic with a smooth surface and spindle-shaped stomatal pore in *Suregada multiflora* but an oval-shaped stomatal pore and circular stomata in *Suregada glomerulata*. The largest size anomocytic stomata with a rough surface was found in *Manihot esculenta*. The medium-sized stomata were surrounded by few and horizontally arranged cuticular striations in *Aleurites moluccana*. The largest sized stomata with high density and thick cuticular striation were found in *Hevea brasiliensis*. The oriented and medium-sized paracytic stomata with a smooth surface and elliptical-shaped stomatal pore were found in all three individuals of *Codiaeum variegatum*. The paracytic but sunken stomata with horizontally arranged cuticular striation were found in *Endospermum chinense*. The same stomatal structure was found in both samples of *Jatropha integerrima* except for they were rich density and oriented stomata with thick and horizontally arranged cuticular striation. Similarly, the sunken stomata with reticulate cuticular striation were observed for *Vernicia montana*. In *Croton flavens* no stomata were observed due to the thick covering of lepidote trichomes.

Similarly, in Phyllanthaceae the anisocytic mixed with paracytic stomata with low density was found in *Antidesma bunius* and *A. montanum* (II). The oval-shaped anisocytic stomata have spindle-shaped stomatal pores and rough surfaces in *Baccaurea ramiflora*. *The oriented and somewhat arrested diacytic and anomocytic stomata* with smooth surfaces *were found in both individuals of Bischofia javanica. The small size and aborted* anomocytic type of stomata with spindle-shaped stomatal pores were noted in *Glochidion sphaerogynum* and *G. zeylanicum*. However, the stoma was not aborted in the former species. The paracytic type was found in *Bridelia balansae*, *Bridellia tomentosa*, *Glochidion wrightii*, *Aporosa dioca*,

*Breynia rostrata*, and *Cleistanthus sumatranus*. However, in *Bridelia balansae* and *B. tomentosa* the stomata with less density and densely papillose surfaces were analyzed and that can easily be differentiated from the rest of the species. The largest size, arrested, sunken, and oval-shaped stomatal pore having less density at the abaxial surface was found in *Glochidion wrightii*, oriented stomata with a smooth surface in *Aporosa dioca*, large size and oriented with a papillose surface in *Breynia rostrata*, while the aborted, appressed, and small-sized stomata were found in *Cleistanthus sumatranus* (Table 3).

**Table 3.** The foliar epidermal traits of the Euphorbiaceae and Phyllanthaceae taxa.

| Species | Epidermal Cell Shape | Anticlinal Wall Pattern | Stomata Type |
|---|---|---|---|
| **Acalyphoideae** | | | |
| *Acalypha hispida* | Polygonal | Straight/sinus | Anomocytic/ Anisocytic |
| *Acalypha indica* | Irregular | sinus | Anomocytic |
| *Acalypha wilkesiana* | Polygonal | Straight | Anisocytic |
| *Alchornea rugosa* | Irregular | Undulate | Anomocytic |
| *Macaranga corymbosa* | Irregular | Sinus | Anomocytic |
| *Mallotus apelta* | Irregular | Deeply undulate | Not visible |
| *Mallotus barbatus* | Irregular | Straight/sinus | Anisocytic |
| *Mallotus paniculatus* | Irregular | Straight/undulate | Paracytic |
| *Ricinus communis* | Irregular/elongated | undulate | Anisocytic |
| **Euphorbioideae** | | | |
| *Euphorbia heterophylla* | Irregular | Sinus | Anomocytic/Anisocytic |
| *Excoecaria cochinchinensis* | Irregular | Deeply undulate | Anisocytic |
| *Hura crepitans* | Irregular | Undulate | Anomocytic |
| *Triadica cochinchinensis* | Irregular | Sinus | Anomocytic |
| *Triadica sebifera* | Polygonal/irregular | Straight | Anomocytic |
| **Crotonoideae** | | | |
| *Aleurites moluccana* | Polygonal/irregular | Straight/sinus | Paracytic |
| *Codiaeum variegatum* (I) | Irregular | Undulate | Paracytic |
| *Codiaeum variegatum* (II) | Irregular | Undulate | Paracytic |
| *Codiaeum variegatum* (III) | Irregular | Deeply undulate | Paracytic |
| *Croton flavens* | Irregular | Undulate | Not visible |
| *Croton hirtus* | Irregular | Slightly wavy | Anomocytic/Anisocytic/ Paracytic |
| *Endospermum chinense* | Polygonal/Irregular | Straight/Sinus | Paracytic |
| *Hevea brasiliensis* | Irregular | Sinus | Anomocytic |
| *Jatropha integerrima* (I) | Polygonal | Straight | Paracytic |
| *Jatropha integerrima* (II) | Polygonal | Straight | Paracytic |
| *Manihot esculenta* | Irregular | Deeply undulate | Anomocytic |
| *Suregada glomerulata* | Polygonal | Sinus | Anomocytic/Paracytic |
| *Suregada multiflora* | Irregular | Undulate | Anomocytic/Paracytic |
| *Vernicia montana* | Irregular | Sinus | Paracytic |
| **Phyllanthaceae** | | | |
| *Antidesma bunius* | Irregular | Undulate | Anisocytic/Paracytic |
| *Antidesma montanum* (I) | Irregular | Sinus | Paracytic |
| *Antidesma montanum* (II) | Irregular | Undulate | Anomocytic/Paracytic |
| *Aporosa dioca* | Irregular | Sinus | Paracytic |
| *Baccaurea ramiflora* | Irregular | Deeply undulate | Anisocytic |
| *Bischofia javanica* (I) | Irregular | Undulate/Sinus | Diacytic/Anomocytic |
| *Bischofia javanica* (II) | Irregular | Undulate | Anomocytic |
| *Breynia rostrata* | Polygonal/irregular | Straight/Sinus | Paracytic |
| *Bridelia balansae* | Polygonal | Straight | Paracytic |
| *Bridellia tomentosa* | Irregular | Sinus | Paracytic |
| *Cleistanthus sumatranus* | Irregular | Undulate | Paracytic |
| *Glochidion sphaerogynum* | Polygonal/Irregular | Straight/Undulate | Anomocytic |
| *Glochidion zeylanicum* | Irregular | Undulate | Anomocytic |
| *Glochidion wrightii* | Irregular | Undulate | Paracytic |

The chord diagram showed variation among the analyzed traits. The interactivity was an advantage in making the chord diagram understandable. The numbers outside the circle represented the number of species. The stomata were found at the adaxial surface

in four species, whereas in about forty species it was found at the abaxial surface. Similarly, at the adaxial surface, the NGTs were found in four species whereas at the abaxial surface it was noted in about sixteen species. The GTs at the adaxial surface were found in nine species whereas in 13 species they were found at the abaxial surface (Figure 14).

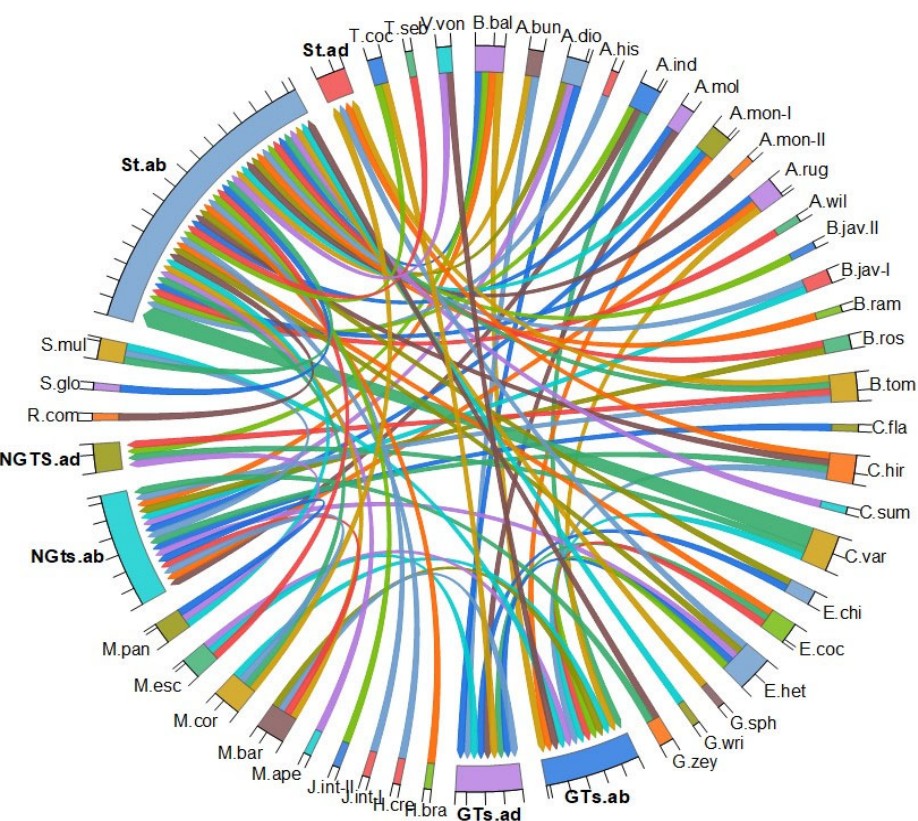

**Figure 14.** Chord diagram showing a correlation between the leaf epidermal traits and species. St.ad, stomata the adaxial surface; St.ab, stomata at the abaxial surface; NGTs.ad, nonglandular trichomes at the adaxial surface; NGTs.ab, nonglandular trichomes at the abaxial surface; GTs.ad, glandular trichomes at the adaxial surface; GTs.ab, glandular trichomes at the abaxial surface.

The multivariate detrended correspondence analysis (DCA) was performed to describe the variation among the Euphorbiaceae taxa. For DCA, the data were arranged first in four different groups (subfamily Acalyphoideae, Crotonoideae, Euphorbioideae, and Phyllanthaceae). Then the variation caused by the epidermis quantitative traits was explained for each group that we set before the analysis. Interval variables were derived from a nominal variable in sets of data and established the canonical correlation with the intervals in a manner that has the variable with the highest multiple correlations first, followed by lesser correlated interval variables. The greater size of the ellipses showed the high variability among the species within the subfamily. The subfamily Euphorbioideae showed high variability followed by Crotonoideae, Phyllanthaceae, and Acalyphoideae. Smaller size ellipses showed a close relationship and less variation between the species of the specific group. For example, the subfamily Acalyphoideae showed less variation among the taxa. The multivariate DCA showed a variation in the eigenvalues that ranged from 0.01388 to 0.33981. The high variability was found in axis 1 with an eigenvalue of 0.3981 followed by axis 2 having a 0.1555 eigenvalue. Similarly, axis 3 had an eigenvalue was 0.04751 and axis 4 had a 0.01388 eigenvalue (Figure 15A,B and Table 2).

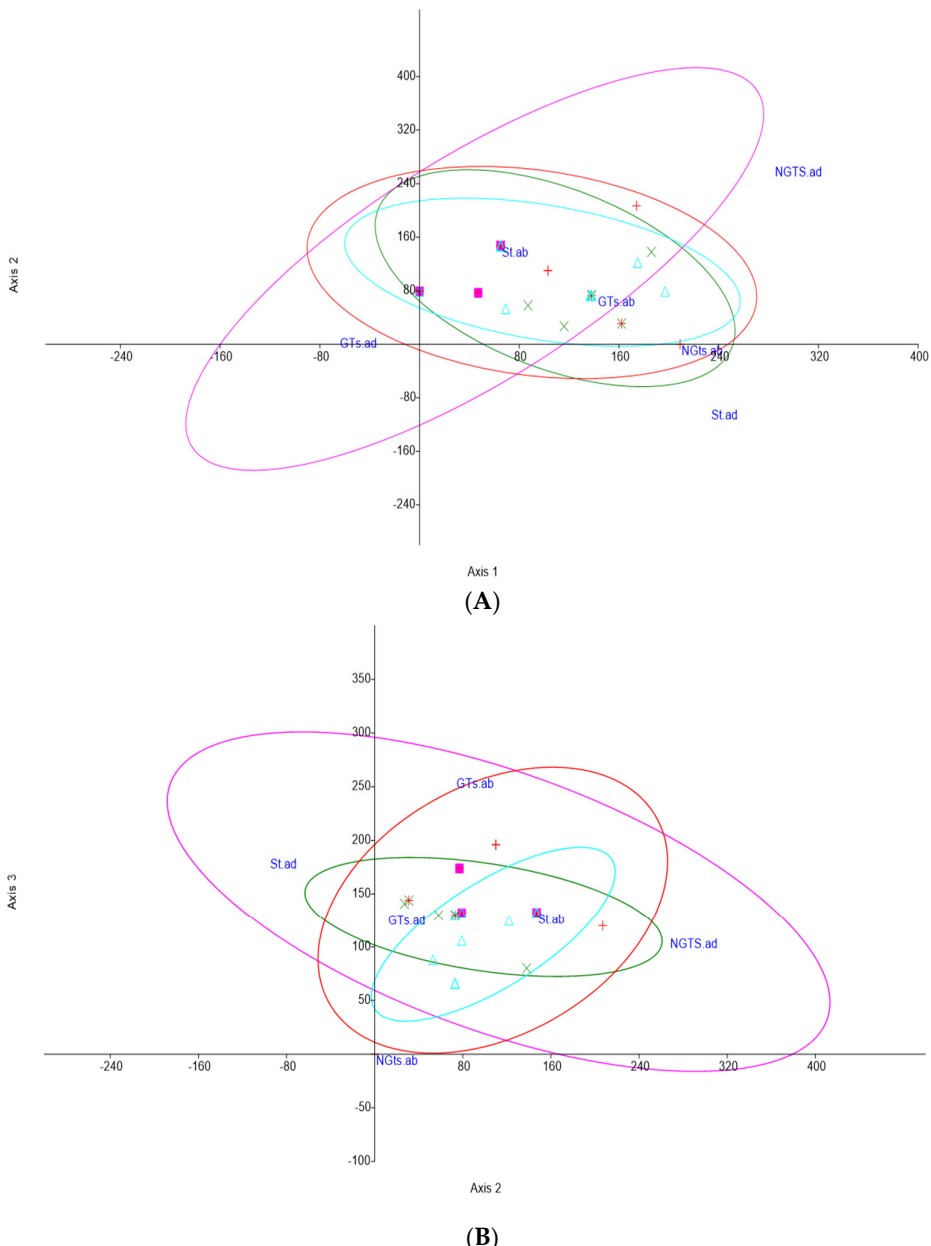

**Figure 15.** (**A**) The detrended correspondence analysis of the presence and absence of leaf epidermal traits of Euphorbiaceae and Phyllanthaceae. Axis 1 has an eigenvalue of 0.3981 and axis 2 has an eigenvalue of 0.1555. (**B**) Axis 3 with an eigenvalue of 0.04751.

Similarly, the principal component analysis was performed by using four variables, which included, stomata length, stomata width, stomatal pore length, and stomatal pore width. As a result, a close and positive relationship was found between stomata length and stomata width. The angle between the vectors of these two traits is very close confirming the close relationship between these two traits of the species. The angle between stomatal pore length and stomatal pore width is larger confirming the weak relationship between these two traits of the analyzed taxa (Figure 16A,B). The PC 1 showed a significant variation (77.73%) while the PC 2 showed a (14.43%) variation, confirming that these traits are taxonomically important and can be used as an additional tool to define species boundaries at various taxonomic levels (Tables 4 and 5).

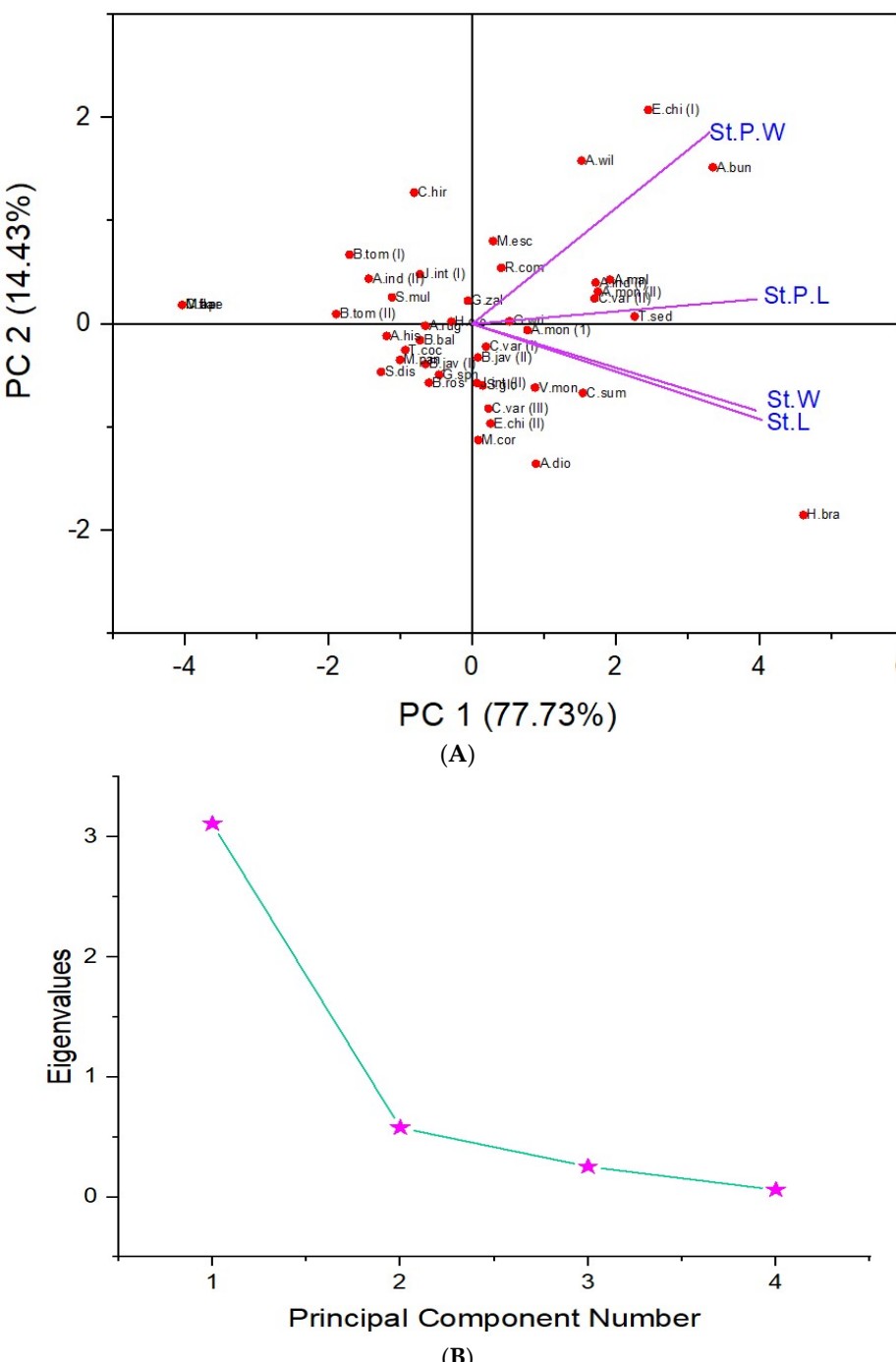

**Figure 16.** (**A**) The principal component analysis of the analyzed taxa based on the stomata length, stomata width, stomatal pore length, and stomatal pore width. The PC 1 showed a significant variation (77.73%), while the PC 2 showed a 14.43% variation. (**B**) A scattered plot of the four principal components.

**Table 4.** Eigenvalues of the correlations matrix.

| PCs | Eigenvalue | Percentage of Variance | Cumulative |
| --- | --- | --- | --- |
| 1 | 3.10932 | 77.73% | 77.73% |
| 2 | 0.57706 | 14.43% | 92.16% |
| 3 | 0.25408 | 6.35% | 98.51% |
| 4 | 0.05955 | 1.49% | 100.00% |

**Table 5.** Score data of the four PCs based on the stomata length and width and stomatal pore length and width.

| Species | PC-1 (77.73%) | PC-2 (14.43%) | PC-3 (6.35%) | PC-4 (1.49%) |
| --- | --- | --- | --- | --- |
| A.wil | 1.52312 | 1.58072 | −0.66743 | 0.21249 |
| A.rug | −0.65164 | −0.01798 | −0.25137 | −0.26137 |
| M.ape | −4.03681 | 0.18294 | 0.04889 | −0.05892 |
| M.pan | −1.00382 | −0.3487 | 0.35307 | −0.05137 |
| R.com | 0.40321 | 0.54186 | −0.40497 | 0.2036 |
| A.ind (I) | 1.71856 | 0.39953 | −0.45165 | 0.10073 |
| A.his | −1.19026 | −0.11769 | −0.17539 | −0.37122 |
| M.cor | 0.08357 | −1.1263 | 0.37629 | 0.16677 |
| M.bar | −4.03681 | 0.18294 | 0.04889 | −0.05892 |
| A.ind (II) | −1.43538 | 0.43737 | −0.17877 | 0.06437 |
| C.var (I) | 0.19273 | −0.21967 | 0.76075 | 0.37488 |
| J.int (I) | −0.73004 | 0.48071 | −0.13033 | −0.08692 |
| S.mul | −1.11482 | 0.25724 | −0.26761 | 0.09307 |
| E.chi (I) | 2.45294 | 2.0738 | 0.2234 | 0.08971 |
| C.var (II) | 1.70031 | 0.24545 | 0.68709 | −0.48597 |
| C.var (III) | 0.22842 | −0.82149 | 0.87356 | −0.17897 |
| H.bra | 4.61258 | −1.85297 | −1.12377 | −0.14421 |
| J.int (II) | 0.06966 | −0.57359 | 0.00296 | −0.28261 |
| E.chi (II) | 0.25567 | −0.96533 | −0.01172 | −0.05042 |
| A.mal | 1.91674 | 0.42808 | 0.92577 | 0.44469 |
| S.glo | 0.14414 | −0.59591 | 0.92228 | 0.16958 |
| M.esc | 0.29184 | 0.80224 | −0.74276 | −0.01195 |
| C.hir | −0.80823 | 1.27168 | 0.63454 | −0.12447 |
| C.fla | −4.03681 | 0.18294 | 0.04889 | −0.05892 |
| V.mon | 0.87495 | −0.61808 | 0.75408 | −0.08145 |
| S.dis | −1.26875 | −0.46528 | −0.42473 | 0.16515 |
| H.cre | −0.29005 | 0.02119 | −0.10468 | 0.16743 |
| T.sed | 2.26342 | 0.0724 | 0.15401 | −0.66746 |
| T.coc | −0.93156 | −0.25398 | −0.30908 | −0.24724 |
| A.dio | 0.8871 | −1.35552 | 0.40362 | 0.47264 |
| B.ros | −0.60293 | −0.57018 | −0.1199 | −0.07701 |
| C.sum | 1.54006 | −0.67051 | −0.64211 | 0.35876 |
| B.jav (I) | −0.64835 | −0.38895 | 0.04582 | 0.16382 |
| B.bal | −0.72521 | −0.15805 | 0.38216 | 0.12298 |
| A.bun | 3.35165 | 1.51595 | 0.22418 | 0.16267 |
| G.wri | 0.51657 | 0.02581 | −0.33469 | −0.32235 |
| G.sph | −0.45826 | −0.49411 | −0.62705 | 0.3222 |
| A.mon (1) | 0.77023 | −0.06024 | 0.72106 | −0.11255 |
| B.jav (II) | 0.07897 | −0.32685 | −0.70857 | 0.04106 |
| B.tom (I) | −1.7074 | 0.66884 | −0.26058 | 0.13338 |
| B.tom (II) | −1.89203 | 0.09449 | −0.39259 | 0.17135 |
| G.zal | −0.06004 | 0.22395 | −0.38608 | −0.25997 |
| A.mon (II) | 1.75277 | 0.31122 | 0.12455 | −0.20706 |

Figure 17 shows a cluster heat map of the analyzed taxa of the Euphorbiaceae and Phyllanthaceae based on the leaf epidermal traits that include stomata length, stomata width, stomatal pore length, and stomatal pore width.

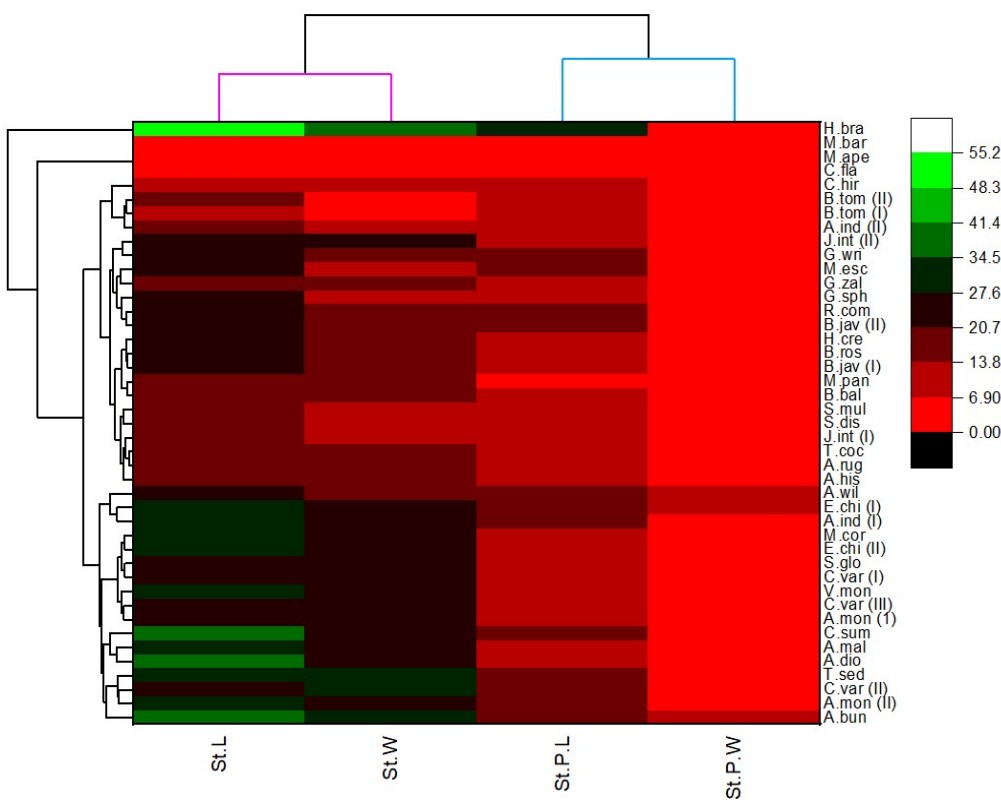

**Figure 17.** A cluster heat map of the analyzed taxa based on the leaf epidermal traits: St.L, stomata length; St.W, stomata width; St.P.L, stomatal pore length; and St.P.W, stomatal pore width.

The box plot shows the distribution of variables of the Euphorbiaceae and Phyllanthaceae taxa. The dot inside the rectangle is the median, the rectangle shows 50% of interquartile range, the ends show the amplitude variation based on the four-leaf anatomical traits (Figure 18).

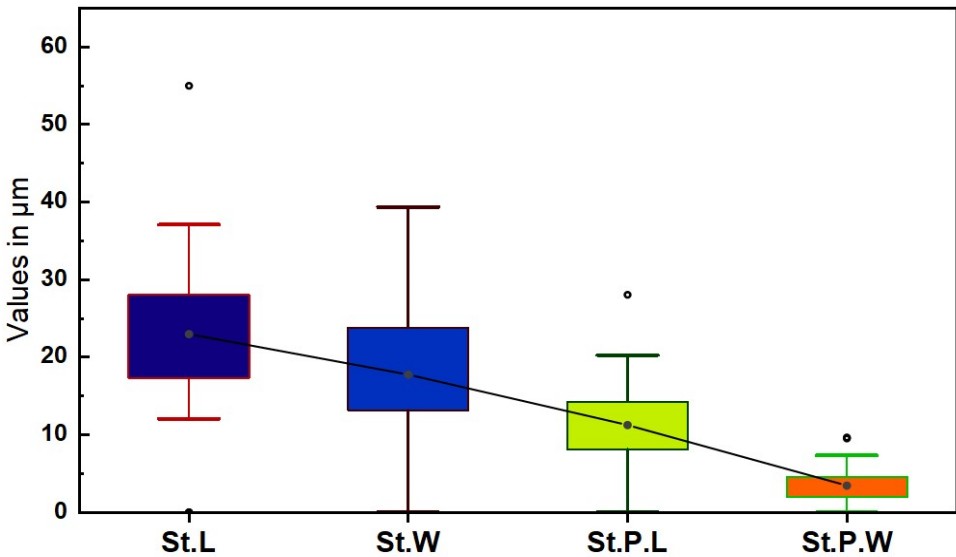

**Figure 18.** Box plot showing the distribution of variables of the Euphorbiaceae and Phyllanthaceae taxa. The dot inside the rectangle is the median, the rectangle shows 50% of interquartile range, the ends show the amplitude variation. St.L, stomata length; St.W, stomata width; St.P.L, stomatal pore length; and St.P., width.

## 4. Discussion

This study represents the most comprehensive investigation of the foliar epidermal anatomy of the family Euphorbiaceae (Acalyphoideae, Euphorbioideae, and Crotonoideae) and Phyllanthaceae. The analyzed traits proposed here highlight the significance of these characters in the systematic of the family. Foliar epidermal anatomy reveals many taxonomic characteristics that can be used in the delimitation of genera, tribes, or species in various families of Angiosperms [11]. The leaf anatomical studies aiming to indicate specific diagnostic characteristics of the Euphorbiaceae and Phyllanthaceae taxa collected from different geographical regions of Hainan Island in South China are still not documented. Our data confirmed the usefulness of the leaf anatomical trait in the taxonomic delimitations of the Euphorbiaceae taxa.

Different types of trichomes were observed in various taxa of the Euphorbiaceae. Selected light and SEM micrographs of the foliar epidermal were illustrated in Figures 1–13. In general, two main types of trichomes were observed on both the adaxial and abaxial surface of the leaf, nonglandular trichomes (NGTs) and glandular trichomes (GTs). The NGTs were further classified into long and short that including short conical and long falcate shapes. Glandular trichomes (GTs) can be divided into capitate, subsessile capitate, sessile capitate, peltate, and lepidote-subentire GTs. The most common type of trichomes in the investigated taxa was short NGTs and long NGTs as well as short, stalked capitate glandular trichomes (GTs). Both NGTs and GTs were mostly distributed at the abaxial surface.

### Crotonoideae

In *Croton hirtus*, highly branched stellate NGTs with arms ranging 8–13 were observed with high density at the abaxial surface. The simple stellate trichomes with 1–12 radiates are reported in Nigerian *Croton hirtus* [65], which supports our findings except for the number of radiates. The small papilose NGTs were occasionally found in *Croton flavens* at the adaxial surface. The lepidote-subentire NGTs with an oriented center were found in *Croton flavens* and covered the entire abaxial surface while infrequently distributed at the adaxial surface and should be considered as a potential taxonomic trait for the classification of the species. These findings are in conjunction with Webster [66], whoe

also reported that trichomes may vary from lipidote in Cuban *C.myricifolius* to lipidote with an erect central radius in *C. brittonianus* to stellate in *C. bispinosus*. A taxonomic review of the genus *Croton* (Euphorbiaceae) has been investigated previously [67], and they reported various types of trichomes ranging from uniform lepidote to lipidote with an erect central radius to stellate trichomes that have a significant taxonomic value to delimit and define species boundaries within subfamily Crotonoideae. The *Croton flavens* share general morphology with most of the species throughout *Croton* section *Adenophylli* which has complicated the application of names to these species [68]. Hence, micromorphological traits will help to solve this taxonomic problem. Various types of NGTs are frequently mentioned and recognized as relevant taxonomic characters for *Croton* constituting important elements in the recognition of the genus and the characterization of the taxa [69]. In the plants analyzed here, we found that the different morphologies of nonsecreting trichomes could be applied in their recognition, corroborating the proposals of other authors who also point out the use of these appendages in the delimitation of clades or phylogenetic approaches [70,71]. A previous study about the comparative anatomy of Croton [69] or Crotoneae [12] demonstrated that types of trichomes are useful for taxonomy mostly at a species level. On the other hand, we noted trichome morphology seemed to be plastic among the studied species, sometimes even within the same individual. For example, in all three individuals of *Codiaeum variegatum*, only one had sessile capitate GTs. Thus, trichome features should be used with caution for taxonomic purposes potentially in all the Crotonoideae taxa [72].

In *Croton hirtus* both abaxial and adaxial epidermis surfaces of the leaf contained paracytic, anomocytic, and anisocytic types of stomata but differed in the presence of tetracytic stomata on the abaxial surface. The epidermal cell shape was irregular, and the anticlinal wall pattern was slightly wavy on both epidermal surfaces [33]. These traits were corroborated with our results except for the tetracytic stomata and irregular-shaped epidermal cells in *C. hirtus*. In comparison with other species, the *C. lobatus* had both tetracytic and paracytic stomata on both surfaces but differed in the presence of anisocytic stomata on the abaxial surface while irregular cell shape and a wavy, anticlinal walls were observed on both surfaces. Similarly, in *C. zambesicus* both paracytic and tetracytic stomata were found with a straight to slightly curved anticlinal wall pattern and polygonal epidermal cells [33]. The anomocytic stomata were found in *Hevea brasiliensis*. However, Martins and Zieri, [73] studied the anatomy and morphology of clones of *H. brasiliences* that allowed the clones to differentiate and provided information on the location and disposition of laticifers in the leaf tissues. The leaf anatomy of *Vernicia* and *Aleurites* has been investigated by Han, [27] who reported a paracytic-type stomata in *Aleurites moluccana* and *Vernicia monatana*. In the study, Han found the abaxial surface, paracytic stomata showing cuticular straite ornamentations in *Vernicia montana* and stomatal ledges curved cuticular ridges of peristomatal rim in *Aleurites moluccana* [27] and concluded that these anatomical features can separate the two taxa, which corroborates our findings. The genus *Vernicia* has not been accepted for a long time, but micromorphological features showed additional evidence that the recent separation of two genera by Shaw, [74] is reasonable.

Overall, the paracytic type of stomata predominantly occurs in *Aleurites moluccana*, all three samples of *Codiaeum variegatum*, two samples of *Jatropha integerrima*, *Endospermum chinense*, *Vernicia montana*, and *Suregada multiflora*. The dominance of paracytic stomata among these species was in accordance with the previous study of Dee Sá-Haiad et al., [39] of some closely related species in the tribe Crotoneae. Hence, this suggests that Crotonoideae taxa could be characterized by the dominance of paracytic stomata followed by anomocytic stomata at the leaf epidermis. The discontinuous, sometimes diffuse striations encircling and infrequently radiating from the guard cells in a parallel direction are found in *Jatropha panduruefolia* [8], which were closely related to *Jatropha integerrima*. The foliar anatomy showed that *Jatropha* shares similar traits, yet some of them can be useful in delimiting the studied species [75]. In the present study, the epidermal cell and anticlinal

wall patterns were polygonal to irregular, and straight to wavy, respectively. Together all these traits are useful to define species boundaries within *Crotonoideae*.

### Acalyphoideae

In the subfamily Acalyphoideae, the adaxial surface is overlaid by a thick cuticle with the anticlinal walls mostly straight followed by the sinus, deeply undulate to undulate. The stomatal frequency and the anticlinal wall sinuation are less, whereas the size of the epidermal cells is larger when compared to with abaxial surface. In contrast, the abaxial cells are usually smaller in surface area and with a comparatively thinner cuticle, a higher stomatal frequency, and more pronounced anticlinal wall sinuations than the adaxial surface. The anticlinal wall is mostly straight mixed with sinus in *Acalypha hispida*, *Acalypha wilkesiana*, *Mallotus barbatus*, and *Mallotus paniculatus* and rarely sinus in *Acalypha indica* and *Macaranga corymbosa*. In the leaves of *Acalypha wilkesiana* the stomata are absent all over the adaxial surface [8]. The paracytic type is the characteristic of the tribe Acalypheae [11] and forms the basic type of the family [8], which was corroborated in our study of the *Mallotus paniculatus*. However, the anomocytic and anisocytic have been found our study. Similarly, variations in leaf anatomical studies of five *Macaranga* species were investigated which could be used as an additional tool for species identification [76]. However, further anatomical studies are needed to complete the anatomical study of the genus *Macaranga* as the current study is still insufficient. In the present study, no stomata were found at the adaxial in *Acalypha hispida*, *Acalypha indica*, *Acalypha wilkesiana*, *Ricinus communis*, *Mallotus apelta*, *Mallotus barbatus*, or *Mallotus paniculatus*. At the abaxial surface, the glandular trichome (GTs) was found in *Acalypha indica, Mallotus barbatus,* and *Macaranga corymbosa* which is in accordance with previous findings of Hussin et al., [77] for the *Mallotus taxa*. Furthermore, they also reported NGTs in most of the *Mallotus* species while the peltate trichomes were recommended as a useful diagnostic trait. Solereder [10] suggests that the GTs are more frequent in the Acalyphae than elsewhere in the Euphorbiaceae. In *M. philippensis*, the presence of trichome-lined crypts at the abaxial surface is the diagnostic character of the species. These anatomical characters in combination are very useful in the identification of *Mallotus* species. However, no comprehensive study about the leaf epidermal anatomy of Acalyphoideae has been previously reported. Hence, further studies are needed to explore the leaf anatomical traits and their relevance in the Acalyphoideae taxa.

### Euphorbioideae

The leaf anatomical traits of the *Euphorbia* have a significant taxonomic value. The presence and absence of trichomes were useful traits for the identification of *Euphorbia* taxa [49]. In our study, the NGTs were found in *Euphorbia heterophylla*, while absent in the rest of the species. Similarly, the GTs was found in *Euphorbia heterophylla* and *Excoecaria cochinchinensis*, while absent in the rest of the species. In comparison with other Euphorbia species, the anisocytic stomata and the irregular-shaped epidermal cells are found in the *Euphorbia cotinifolia*, *E. neriifolia*, and *E. hirta* [46], while paracytic are found in *E. hirta* and *E. milii* with polygonal-shaped epidermal cells. However, the anomocytic stomata is more common in *Euhphobia* species [78]. The surface of stomata was papilose in *Triadica cochinchinensis* and should be considered a diagnostic trait of the species. Similarly, the discontinuous and parallel striation radiating from the guard cells was found in *Hura crepitans*. In the present study, the shape of leaf epidermal cells and the anticlinal wall pattern may have also varied among the species. The epidermal cells at the adaxial surface were larger than on the abaxial surface.

### Phyllanthaceae

Various types of stomata were found in the stomata of Phyllanthaceae taxa that include paracytic, anomocytic, anisocytic, and diacytic. The irregular-shaped epidermal cells with paracytic stomata are reported previously in *Bischofia javanica [46]*, which were inconsistent with our findings. We observed the diacytic and anomocytic stomata for both samples of *B. javanica*. Overall, the leaf anatomical studies of the family Phyllanthaceae is

scarce. However, wood anatomy provided a comprehensive survey of the tribe Phyllantheae and showed a systematic relationship [79]. Similarly, the macromorphology of *Glochidion* is analyzed by Yao and Zhang [80] for the reinstatement of three names of the *Gluchidion*. Furthermore, new records of *Glochidion* species from several provinces in China have been reported previously [81]. Hence further studies are still needed to fill the anatomical gap of Gluchidion and explore their taxonomic value. The anisocytic stomata with three unequal sizes of epidermal cells with no particular orientation and nonglandular trichomes (NGTs) were found in *Aporosa* species [82]. However, we found paracytic stomata in *Aporosa dioca*. In Phyllanthaceae, Webster and Ellis (1962) [83] hold that trends of evolution in Phyllanthinae are from the woody to herbaceous condition. The comparison of woody species with paracytic stomata and herbaceous with derived anisocytic stomata lends additional support to the findings of Webster and Ellis [83]. Leaf architecture and epidermal anatomy vary in a taxonomically useful way in the Phyllanthaceae [84]. Anatomical studies are a systematic line of evidence used in combination with other systematic lines to arrive at a good taxonomic condition [85]. Endress et al. [86], dealt extensively with the relative importance of anatomical features in the systematic positioning of various angiosperm groups. Despite this, different authors in various plant groups have studied the scientific importance and specific implication of anatomical features of the Phyllanthaceae taxa [87–89]. Therefore, basic anatomical information provides data that will clarify the confusion in the identity of these taxa.

**Basic stomatal type**

In Euphorbiaceae, the variation met within the predominant paracytic stomatal type. Its variability, distribution, and classification in different species seemed to be more useful for species delimitation in conjunction with other stomatal types, orientation, ornamentation, and striations. Euphorbiaceae has been shown to constitute a good exemplification of the statement of Cutter, [90], that not all the stomata present on the epidermis are good examples of a single type and a large number of stomata should be examined to determine the most prevalent type. The paracytic type of stomata is the basic stomatal type for the family Euphorbiaceae as it is common in most of the species. The works of Essiett et al. [91] in Euphorbia [12] and Bhavana et al. [51] in foliar anatomy in Crotoneae, Olowokudejo [28] in *Jatroppha*, Levin (1986b), systematic foliar morphology of Phyllanthoideae [21], and foliar anatomy of Euphorbiaceae also reported the basic paracytic-type stomata, followed by anomocytic- and anisoytic-type stomata. The anisocytic types are reported by Rothdauscher [92] in *Aporosa* and *Baccaurea*. Similarly, the predominant stomata were anamocytic while the paracytic, anisocytic, and hexacytic are reported in Euphorbiaceae taxa [42]. Thus, three basic types are apparent in the present study. Some authors have investigated the taxonomic role of stomata and other epidermal traits, such as trichomes, epidermal cells, and anticlinal wall patterns, not only in Euphorbiaceae [41] but in certain dicotylodons as well.

However, the incidence of many stomatal types in the same leaf of some taxa of the Acalyphoideae, Euphorbioideae, Crotonoideae, and Phyllanthaceae is likely to raise some misgivings about the systematic value of this trait. However, its potential taxonomic value, if properly analyzed can be demonstrated. Despite the apparent lack of stability in the stomatal pattern, the taxa are now studied and can be delimited based on a combination of epidermal traits. When many traits are considered together with an understanding of their variability, they have significant taxonomic relevance to define species boundaries at various taxonomic ranks.

The trichomes are widely used as one of the important features in the classification and identification of plants. In the present study, a significant variation was found in trichomes morphology with a potential taxonomic value in distinguishing the genus and species of the Euphorbiaceae. Hence, trichomes are widely used as one of the diagnostic features in the classification and identification of plants. The present study showed them to have significant value in distinguishing the genus and species of Euphorbiaceae. Thus,

the trichomes features can be used to strengthen the systematic of Euphorbiaceae and can be used for species identification [92]. A previous study has reported that variation in foliar trichome characters was able to address the taxonomic conflicts of selected taxa in Acanthaceae [93].

The adaxial epidermis is overlaid by a thick cuticle with the anticlinal walls mostly sinus, undulate, deeply undulate, and rarely straight. The stomatal frequency, distribution, and anticlinal wall sinuations were small, whereas the size of the epidermal cells was large when compared with the lower epidermis. In contrast, the abaxial epidermal cells were usually smaller in surface area and with a comparatively thinner cuticle, and had a higher stomatal frequency with more pronounced anticlinal wall sinuations than the upper. The anticlinal walls were rarely straight. Although the diverse micromorphological epidermal and cuticular characters alone are not sufficient for the delimitation of the various infrageneric groups, it is clear from the present study, that together these anatomical features are useful for the identification, delimitation, and confirmation of species relationships. Mueller, [94] and Cutler and Brandham [95] have shown that many leaf surface patterns are under strong genetic control and therefore little affected by the environment. However, similar to all other taxonomic evidence, the anatomical traits must be interpreted with great circumspection.

The chord diagram visualizes the interrelationships between the traits and species. The connection between the traits and species is used to display that they share common traits. For example, the distribution of stomata and various types of trichomes (NGTs and GTs). Hence this makes the diagram ideal for comparing the similarities within a dataset or groups of data. The nodes were arranged along the circle with the relationships between the traits and species to each other through the arcs whereas the values were assigned to each connection which was represented proportionally by the size of each arc. Different colors grouped the data into various categories that aided in making comparisons and distinguishing groups. The multivariate principal component, detrended correspondence analysis, box plot, and cluster heat map were found to be useful for a better understanding of the results and to show the relationship between the species and distribution of the leaf anatomical data of both Euphorbiaceae and Phyllanthaceae from Hainan Island, South China.

## 5. Conclusions

The subfamily, Acalyphoideae, Euphorbioideae, Crotonoideae (Family, Euphorbiaceae) and family Phyllanthaceae showed great morphological variability that is reflected in their anatomical diversity, particularly the presence and absence of both glandular (GTs) and nonglandular trichomes (NGTs). The secretary structures are more common in Euphorbiaceae at the abaxial surface. Similarly, variation was seen in the type and structure of stomata while the paracytic-type stomata predominantly occurred in most of the taxa. Some other types, anisocytic, anamocytic, and diacytic have also been reported. The predominant shape of epidermal cells was irregular and rarely polygonal. Similarly, the anticlinal wall pattern may also vary significantly among the species, although the diverse micromorphological epidermal and cuticular characters alone are not sufficient for the delimitation of the various infrageneric groups. It is clear from the present study, that together with quantitative data, the anatomical features allow the identification, delimitation, and confirmation of species relationships at various levels. The detailed description presented here will provide bases for future studies to define species boundaries and will strengthen the systematics of Euphorbiaceae.

**Author Contributions:** Conceptualization, W.L. and S.B.; methodology, S.B.; software, W.L.; validation, W.L.; formal analysis, S.B.; investigation, S.B. and W.L.; resources, S.B. and W.L.; data curation, M.Y.; writing—original draft preparation, S.B., M.Y., M.A. and U.H.; writing—review and editing, M.A. and U.H.; visualization, W.L.; supervision, W.L.; project administration, W.L.; funding acquisition, W.L. All authors have read and agreed to the published version of the manuscript.

**Funding:** This project was supported by the National Natural Science Foundation of China (31870508, 32171772).

**Institutional Review Board Statement:** Not applicable.

**Data Availability Statement:** Not applicable.

**Acknowledgments:** This research article is part of the Ph.D. Dissertation of the first author. We are thankful to Hainan University for providing the SEM facility and the College of Forestry for providing the LM facility. We are also thankful to our lab mates from the College of Forestry for their kind help, useful suggestions, and providing materials.

**Conflicts of Interest:** The authors declare no conflict of interest.

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
