# Peer review of "Leaf Epidermal Traits of Selected Euphorbiaceae and Phyllanthaceae Taxa of Hainan Island and Their Taxonomic Relevance"

_diversity, doi:10.3390/d14100881_

Round 1

Reviewer 1 Report (Previous Reviewer 2)

Dear Authors and Editors,

I read with great attention the sent corrected version of the article entitled "Leaf epidermal traits of selected Euphorbiaceae and Phyllanthaceae taxa of the Hainan Island and their taxonomic relevance". I certify that in the current version of this work all errors have been corrected. I have only minor comments on the technical side, I have no further comments as to the content of the work, I believe that the amended article may be published.

Minor

Figures 12 and 13: I suggest changing the colour of the scale bars from yellow to black

Author Response

I read with great attention the sent corrected version of the article entitled "Leaf epidermal traits of selected Euphorbiaceae and Phyllanthaceae taxa of the Hainan Island and their taxonomic relevance". I certify that in the current version of this work all errors have been corrected. I have only minor comments on the technical side, I have no further comments as to the content of the work, I believe that the amended article may be published.

Minor

Figures 12 and 13: I suggest changing the colour of the scale bars from yellow to black.

Ans: Thank you very much for your kind response. We changed the color of the scale bars from yellow to black in figures 13 and 13.

Reviewer 2 Report (Previous Reviewer 3)

The author's sufficiently address the concerns that I made in my initial review.  With regard to diversity of Euphorbiaceae and Phyllanthaceae, I think that the diversity of these families in the study area (Hainan Island) should be presented, and not data for entire China.  It still needs to be reviewed by an editor who is fluent in English.  Take, for example, the first sentence of the abstract: Euphorbiaceae and Phyllantaceae is one of the most taxonomically complex and diverse 12 family among the other angiosperm groups. Since there are two families, it should be written: Euphorbiaceae and Phyllantaceae are two of the most taxonomically complex and diverse 12 families among the angiosperms.

Author Response

The author's sufficiently address the concerns that I made in my initial review.  With regard to diversity of Euphorbiaceae and Phyllanthaceae, I think that the diversity of these families in the study area (Hainan Island) should be presented, and not data for entire China.  It still needs to be reviewed by an editor who is fluent in English.  Take, for example, the first sentence of the abstract: Euphorbiaceae and Phyllantaceae is one of the most taxonomically complex and diverse families among the other angiosperm groups. Since there are two families, it should be written: Euphorbiaceae and Phyllantaceae are two of the most taxonomically complex and diverse families among the angiosperms.

Ans: Thank you very much for your precious time and useful comments on our manuscript. We carefully check and corrected the grammatical and typographic mistakes all over the manuscript and highlighted the text green where we make changes. Besides, we arranged all the references in the sequence.

This manuscript is a resubmission of an earlier submission. The following is a list of the peer review reports and author responses from that submission.

Round 1

Reviewer 1 Report

               The work represents an important contribution to the knowledge of aspects of the leaf epidermis of representatives of two expressive families of angiosperms (Euphorbiaceae and Phyllanthaceae). However, I point out several suggestions and recommendations throughout the text; the first recommendation is to change the title, given that practically two decades ago the subfamily Phyllanthoideae was dismembered from Euphorbiaceae, constituting a separate family (Phyllanthaceae).

.

Reviewer 2 Report

The Euphorbiaceae family is a taxonomically very difficult systematic group, mainly due to the high phenotypic plasticity of the species. Currently, in the taxonomy of this group, we successfully use molecular tools that allow to verify the species described only on the basis of morphological features. I read the article submitted to Diversity with great attention and interest as I am interested in the taxonomy and systematics of the Euphorbiaceae family. I have already been a reviewer of this manuscript sent to another journal of MDPI, and I see that the version sent now has not been corrected, it still contains very many serious errors.

The methodology used in the article raises the greatest doubts. We know nothing about how many samples the authors analyzed. Are the presented data statistically significant? Were these samples random? The Figures with SEM photographs presented only confirm the belief that the authors do not know the methodology of taxonomic research. The same leaf fragment from different taxa should always be selected for analysis and comparison. Photographs document different leaf fragments! Figures should be supplemented with detailed descriptions, it is not known what part of the leaf surface they represent and adds scale bars. The photographs of SEM once show a fragment of the abaxial epidermis, next to the adaxial epidermis - this is unacceptable (see figs. 12 and 13). In Figure 5d, cells are collapsed and deformed, indicating preparation errors. Figure 10 shows the trichomes, the ultrastructure of which should be described in detail, I suggest choosing other photographs as well, the hairs shown are damaged.  The results of the micromorphological studies presented by the authors, contrary to their opinion, do not allow for species identification. Please remember that Euphorbiaceae is a family with very high phenotypic variability! In this type of research, it is necessary to determine the variability of features on a large sample before assessing their usefulness in the taxonomy. 

Dear Authors, please correct the errors in this manuscript, instead of sending it unchanged manuscript to various journals of MDPI.

Reviewer 3 Report

     This article presents the results of an anatomical study of the Euphorbiaceae species occurring on the island of Hainan, China.  It presents novel data that will be useful for further studies on Euphorbiaceae. I recommend publication if the following comments can be addressed:

There is a major problem with the title.  It should read “uni- and bi-ovulate Euphorbiaceae.”  This problem is repeated in the text. However, the bi-ovulate subfamilies are no longer included in Euphorbiaceae. The subfamily Phyllanthoideae has for the past 15 years been universally accepted as distinct from Euphorbiaceae, for example by the Angiosperm Phylogeny Group where many references supporting this can be found.  It is now treated as Phyllanthaceae.  This updated classification should be followed.

In line 111, the authors mention that one of the objectives is to “to explore the diversity of Euphorbiaceae taxa”.  But in what aspect? Anatomically?

The authors cite (Chen et al., 2016) concerning the diversity of of Hainan Island.  They should also mention something about the diversity of Euphorbiaceae and Phyllanthaceae on the Island.  How many species are there and what percentage of genera and species were sampled in this study.

In anatomical studies such as this, it is important to cite vouchers specimens, and without these that data will always be questionable. I was taught “no voucher, no data.”

Many of the taxa that they study are introduced, for example, Hevea brasiliensis and Manihot esculenta. The authors should mention which species are introduced and which species are native.

In Table 1, there is some inconsistency regarding the authors, for example, “L.” vs. “Linn.” and “Müll.Arg.” vs. “Müller Argoviensis”. Also, some authors are not included, and others are underlined.  This should be corrected.

The English is perfectly understandable in most parts of the text, but there are many minor errors and awkward constructions.  The manuscript should undergo an English review.

Some specific minor comments include:

Line 53: Euphorbia should be in italics.

Lines 55–56: cite references for the statement: Morphological and Phylogenetic studies have helped to clarify the systematic of complex groups, i.e. tribe Crotoneae. Also, “phylogenetic” should be lowercase.

Line 96: “and Vernicia.” Something seems to be missing.  What is it about Vernicia that should be mentioned.

Line 124: 1000 to 1200 mm of rainfall a year doesn’t sound too dry to me.

Line 124: change “12000 mm” to “1200 mm”

Lines 130–134: Why are these in red?

Lines 230–236: Why are these in italics?

Line 287: Remove italics from the words that are not scientific names.

Lines 416–419.  Only the scientific name should be italicized.

Line 624: “Vernicia monatana” change to “Vernicia montana”.

Line 625: “Vernicia Montana” change to “Vernicia montana”.